# Large Scale Empirical Bayesian Causal Discovery Using Total Effect Estimates From Intervention Data

## Abstract

Inferring the causal relationships among a set of variables in the form of a directed acyclic graph (DAG) is an important but notoriously challenging problem. Recently, advancements in high-throughput genomic perturbation screens have inspired the development of methods that leverage interventional data to improve model identification. However, existing methods still suffer from poor performance on large-scale tasks and fail to quantify uncertainty. Here, we propose Interventional Bayesian Causal Discovery (IBCD), an empirical Bayesian framework that infers the causal graph by using intervention data to estimate the effect of each variable on every other, then inferring the posterior graph given these estimates. For tractability, our approach models the likelihood of the matrix of estimated total causal effects, which can be approximated by a matrix normal distribution, rather than the full data matrix. We place a spike-and-slab horseshoe prior on the edges and separately learn data-driven weights for scale-free and Erdős–Rényi structures from observational data, treating each edge as a latent variable to enable uncertainty-aware inference. Through extensive simulation, we show that IBCD achieves superior structure recovery compared to existing baselines. We apply IBCD to CRISPR perturbation (Perturb-seq) data on 521 genes, demonstrating that edge posterior inclusion probabilities enable identification of robust graph structures.

## 1 Introduction

Inferring the causal relationships among a set of measured variables is one of the fundamental challenges of science (Bunge, 2012). These relationships are typically represented in a directed acyclic graph (DAG), with nodes representing variables and the presence of an edge from one variable to another indicating the former has a direct causal effect on the latter. Unfortunately the DAG is not identifiable from observational data alone (Verma & Pearl, 1990), and identifying the equivalence class of DAGs consistent with the data is NP-hard, making it intractable for problems involving even a modest number of variables (Chickering).

This issue has recently received substantial attention in genomics, where progress in understanding biological mechanisms of disease is now thought to hinge on understanding gene regulatory networks containing thousands of variables (Boyle et al., 2017; Liu et al., 2019; Wray et al., 2018). Fortunately, advances in CRISPR screening techniques such as Perturb-seq (Dixit et al., 2016; Gasperini et al., 2019; Montalbano et al., 2017) allow researchers to systematically perturb specific genes and measure the response, yielding *interventional* data for hundreds to thousands of genes measured in upwards of one-million cells (Replogle et al., 2022; Feng et al., 2024). This has created an ideal setting for the development of methods for inferring causal graphs at scale (Xue et al., 2023; Yang et al., 2018; Brown et al., 2025; Nazaret et al., 2023; Annadani et al., 2023; Hägele et al., 2023; Weinstock et al., 2024).

While differentiable causal discovery methods (Zheng et al., 2018; Brouillard et al., 2020; Nazaret et al., 2023; Lopez et al., 2022) have made tremendous advancements and received substantial attention, they are limited to returning a single optimal DAG without uncertainty quantification. In contrast, Bayesian methods (Madigan et al., 1995; Geiger & Heckerman, 2002; Friedman & Koller, 2003; Agrawal et al., 2018; Hägele et al., 2023; Annadani et al., 2023; Weinstock et al., 2024) aim to characterize the posterior distribution over possible graphs. This is critical as there may be multiple different near-optimal DAGs. Uncertainty

is also invaluable for scientific applications where assessing the degree of confidence in an edge or graph structure is required in order to prioritize follow-up investigations or experiments.

Due to the exponential scaling of the number of DAGs, exact posterior calculation is intractable for all but the smallest problems (Tian & He, 2012). Most practical approaches have therefore focused on using Markov Chain Monte Carlo (MCMC) to sample from the posterior (Geiger & Heckerman, 2002; Friedman & Koller, 2003; Agrawal et al., 2018), however these approaches do not accommodate intervention data. Recent work uses gradient information to further speed up inference (Annadani et al., 2023; Lorch et al., 2021; Cundy et al., 2021; Deleu et al., 2022; Toth et al., 2022; 2024), but these approaches still fail to scale to the volume of data we consider here.

We take an alternative approach. Rather than directly modeling the distribution of the DAG structure or direct edge weights given the data, we use the data to calculate the total causal effect (TCE) of every feature on every other. This reduces the amount of data that needs to be considered when calculating the likelihood. We then approximate the posterior distribution of the direct effect matrix given the total effect matrix. While modeling direct effects from total effects has been previously considered (Hyttinen et al., 2012; Brown et al., 2025; Weinstock et al., 2024), we present the first fully Bayesian treatment of this model and the first causal discovery method that scales to large problems while quantifying uncertainty. Moreover, while recent causal discovery approaches often model interventions, they typically assume *hard* or *structural* interventions which alter the DAG. We instead consider more realistic *soft* interventions, understood as changes to causal mechanisms that preserve the graph structure. In our model, these *soft* interventions take the form of *shift* interventions (Sani et al., 2020), implemented as additive perturbations to the targeted variables that leave the structure intact. We summarize our contributions as follows:

- We formulate the problem of inferring the DAG as finding the inverse of the TCE matrix in the linear-autoregressive model using soft interventions.

- We show that asymptotically normal estimators of the TCE entries lead to a sampling distribution that can be well-approximated by a matrix normal distribution. This substantially reduces the dimensionality of the sampling covariance and enables efficient likelihood-based inference.

- We propose a hybrid empirical Bayesian prior over the DAG by combining global sparsity patterns with edge specific weights optimized from covariance estimates in the observational data. We learn Erdős–Rényi (ER) and scale-free (SF) priors separately through structure-specific optimization, yielding a flexible, data-adaptive prior.

- We perform extensive empirical analyses while varying properties of the DAG, showing superior performance to existing approaches along with the ability to quantify edge inclusion uncertainty.

- We assess uncertainty on real data by 1) performing a cross-validation analysis of 521 genes in the "essential" Perturb-seq screen (Replogle et al., 2022) and 2) performing an out of distribution replication analysis between the "essential" and "genome-wide" Perturb-seq screens, showing that posterior inclusion probabilities (PIPs) provide robust and practical measures of edge significance across folds and across datasets.

## 2 Background

### 2.1 Causal discovery

Causal discovery is the process of learning a causal graphical model (CGM), specifically a structural equation model (SEM), on a set of variables from data (Pearl, 2009). The model is represented by a DAG $\mathcal{G} = (V, E)$ where each node $i \in V$ corresponds to one of $D$ variables $\{y_i\}_{i=1}^D$ and an edge $(i, j) \in E$ represents a direct effect of $y_i$ on $y_j$. This is augmented by a set of functions detailing the relationship between each variable and its causal parents, $y_i = f_i(y_{pa_i^{\mathcal{G}}}, \epsilon_i)$, where $\epsilon_i$ is an exogenous noise variable. Typically the variables $\{\epsilon_i\}_{i=1}^D$ are assumed random, unmeasured and independent, and we can equivalently specify the model in terms of

probability distributions $p_i(y_i|y_{pa_i^\mathcal{G}})$. Assuming causal sufficiency, *i.e.* that all other relevant variables are measured, this gives the causal model.

The data, $\mathcal{D}$, can be either purely observational, consisting of $N$ measurements of the variables, $Y = [y_{ni}]_{n=1,i=1}^{N,D}$, or interventional. With interventional data, the measured variables are augmented by a binary indicator matrix $X = [x_{nm}]_{n=1,m=1}^{N,M}$, where $x_{nm} = 1$ if sample $n$ received intervention $m$. These interventions can be either *hard* or *soft*. With hard interventions, incoming edges to the intervened-on variables are assumed to be broken, yielding a set of interventional distributions $p_{m_i}(x_i)$ for each $i \in \mathcal{I}_m$. With soft interventions, the distribution is changed without breaking the incoming connections, $p_{m_i}(x_i|y_{pa_i^G})$. The targets of the interventions $\{\mathcal{I}_m\}$ can be assumed to be known or unknown.

## 2.2 The linear-autoregressive causal model

While inferring the parents of each node $pa_i^\mathcal{G}$ and the full form of $p_i$ for each $y_i$ is an admirable goal, we instead choose to summarize the direct effect of one node on another with an average causal effect. Therefore, we assume a linear structural causal model, $f_j(y_{pa_j^\mathcal{G}}, \epsilon_i) := \sum_i y_i G_{ij} + \epsilon_i$, where $G_{ij}$ is the direct effect size of $i$ on $j$. Unlike nonlinear models, where total effects depend on the conditioning point, the linear model yields a single globally defined scalar effect $G_{ij}$. This gives the well-studied linear-autoregressive causal model that we consider here (Geiger & Heckerman, 2002; Hyttinen et al., 2012; Zheng et al., 2018; Brown et al., 2025; Weinstock et al., 2024),

$$Y = YG + \epsilon, \tag{1}$$

with $G = [G_{ij}]_{i=1,j=1}^{D,D}$ the weighted adjacency matrix encoding the graph. We adopt this assumption because linear mechanisms admit identifiable total-effect matrices under complete interventions and remain computationally tractable in large $D$ regime typical of genomic data.

## 2.3 Bayesian causal discovery

In Bayesian causal discovery, the goal is to estimate the entire posterior distribution of CGMs given the data, $p(\mathcal{G}, \Theta|\mathcal{D})$, where $\Theta$ parameterizes the conditional distributions $p_i$ (Friedman & Koller, 2003). This requires prior distributions over DAGs $p(\mathcal{G})$ and parameters $p(\Theta|\mathcal{G})$, and a likelihood $p(\mathcal{D}|\mathcal{G}, \Theta)$, so that Bayes rule can be applied to yield the posterior. Under the linear auto-regressive model the prior and likelihood can be expressed in terms of the weighted adjacency matrix,

$$p(G|\mathcal{D}) \propto p(\mathcal{D}|G)p(G). \tag{2}$$

The posterior encapsulates epistemic uncertainty regarding the structure of the graph given the data. Due to the sheer number of possible graph structures in the posterior, we focus here on summarizing the posterior through the posterior mean graph, which we define as the average weighted adjacency matrix $\mathbb{E}[G|\mathcal{D}]$, and the posterior inclusion probability (PIP) of each edge,

$$\text{PIP}_{ij} = \Pr(|G_{ij}| > \varepsilon \,|\, \mathcal{D}), \tag{3}$$

where $\varepsilon$ is a small edge–existence threshold.

# 3 Interventional Bayesian Causal Discovery

## 3.1 Data model

We augment the standard linear-autoregressive model equation 1 with soft interventions,

$$Y = YG + X\beta + \epsilon, \tag{4}$$

where Y and X are the $N \times D$ and $N \times M$ data and intervention design matrices as before, $\beta$ is the $M \times D$ matrix of unknown intervention on gene effects, and $\epsilon$ is an $N \times D$ exogenous noise matrix.

We assume that the direct targets of the interventions are known but that the effects of the interventions are unknown. That is, we know which entries of $\beta$ are non-zero but not their values. This is a reasonable assumption in Perturb-seq data where the CRISPR guide target is known and off-target effects are minimal (Replogle et al., 2022). We also assume that every feature has at least one instrument, allowing us to obtain $\hat{R}$, the estimated total causal effect matrix. We assume that the exogenous variables $\epsilon_i$ are independent with bounded variance, but make no other assumptions about their distributions $p_{\epsilon_i}$. $R$, true total causal effect matrix, can also be written recursively,

$$R = \left[\sum_k R_{ik}G_{kj}\right]_{i=1,j=1}^{D,D} = \sum_{d=0}^{\max\{|p|,p\in\mathcal{P}^G\}} G^d = (I-G)^{-1} \tag{5}$$

This together with collecting terms of equation 4 and multiplying by the inverse yields the model,

$$Y = (X\beta + \epsilon)(I-G)^{-1} = X\beta R + \epsilon R. \tag{6}$$

This model implies that under complete interventions and in the infinite-sample limit, the graph is identifiable (Hyttinen et al., 2012).

## 3.2 Instrumental variables and soft interventions

When interventions are assumed to be hard, the interventional distribution is typically specified by removing the dependency of the intervened node on its parents, $p_{m_i} = p_{\epsilon_i}$, or by fixing it to a specific value such as 0, $do(y_i = 0)$. If the intervention instead results in a weak perturbation to its target, the interventional distribution is more difficult to specify, but is commonly modeled as a mean shift (Sani et al., 2020). Alternatively, we can treat the interventions as instrumental variables (IVs). $x$ is a valid IV for $y_i$ on $y_j$ if $x$ is not independent of $y_i$ and if $x$ is independent of all exogenous factors affecting $y_j$ given $do(y_i)$ (Pearl, 2009). Thus in our model an intervention $\mathcal{I}_m$ targeting variable $y_m$ is an IV for $y_m$ on every other variable $y_{i\neq m}$.

While there are many techniques for instrument variables analysis (Levis et al., 2024; Imbens, 2014; Baiocchi et al., 2014), the simplest to consider in the linear case given here is two-stage least squares (2SLS). In 2SLS, the average causal effect $\mathbb{E}[y_j|do(y_i)] = R_{ij}y_i$ is estimated by regressing $y_i$ on the instrument(s) $x$ to get $\hat{y}_i$, then regressing $y_j$ on $\hat{y}_i$,

$$\hat{R}_{ij} = (Y_i^\top P_x Y_i)^{-1} Y_i^\top P_x Y_j. \tag{7}$$

with $P_x = x(x^\top x)^{-1}x^\top$ the projection matrix onto the column space of $x$. Note that this measures the *total* causal effect of $y_i$ on $y_j$ under the model $G$, not the direct effect. Letting $\mathcal{P}_{i\to j}^G$ be the set of paths connecting $y_i$ and $y_j$ in $G$, we can write $R_{ij}$ as the sum of the product of the edge values in each path (Hyttinen et al., 2012)

$$R_{ij} = \sum_{p\in\mathcal{P}_{i\to j}^G} \prod_{(k,l)\in p} G_{kl}. \tag{8}$$

**Hard interventions.** While our focus is on soft interventions, we note that the same pipeline can accommodate hard interventions. For an intervention $\mathcal{I}_m$ with target $y_i$, form $G^{(m)}$ by zeroing column $m$ of $G$ and write $Y^{(m)} = Y^{(m)}G^{(m)} + c^{(m)} + \varepsilon^{(m)}$. In this regime $y_i$ is exogenous, so the total effect on $y_j$ is identified from the intervened samples by ordinary least squares, $\hat{R}_{ij} = \left(Y_i^{(m)\top}Y_i^{(m)}\right)^{-1}Y_i^{(m)\top}Y_j^{(m)}$.

## 3.3 Empirical Bayesian model

Our approach is inspired by a common strategy in genome-wide association studies, where it is typically more computationally efficient to work with summary statistics of the data, effect sizes and their standard errors, rather than the full data matrix (Pasaniuc & Price, 2017). Thus, instead of modeling $Y$, we use $Y$ and $X$ to estimate the total effects $\hat{R}_{ij}$ and their covariances $\hat{S}_{ijkl} = \text{Cov}(\hat{R}_{ij}, \hat{R}_{kl})$. We then model $\hat{R}$ with a matrix normal likelihood and place a horseshoe mixture prior on $G$. See Figure 1 for a plate diagram.

**Matrix normal likelihood.** Under standard instrumental variable assumptions, the 2SLS estimator $\hat{R}$ has a normal distribution asymptotically. Accordingly, the likelihood $p\left(\hat{R} \mid G, \hat{S}\right)$ can be approximated by a matrix normal $\mathcal{MN}(R, U, V)$, where the mean $R = (I - G)^{-1}$ represents the true total-effect matrix.. This approximation is motivated by the nearly separable row/column structure of the estimator's covariance, which allows us to model the full covariance matrix $\hat{S}$ more efficiently. Recall that $X \sim \mathcal{MN}_{n \times p}(M, U, V)$, if $\text{vec}(X) \sim \mathcal{N}_{n \cdot p}(\text{vec}(M), V \otimes U)$, where $\otimes$ denotes the Kronecker product. Here, $U \in \mathbb{R}^{n \times n}$ captures row-wise correlations, $V \in \mathbb{R}^{p \times p}$ captures column-wise correlations, and $S = V \otimes U$ defines the full $np \times np$ covariance matrix. In our model, $U$ captures the covariances $\hat{S}_{i \cdot k \cdot}$ between interventions on variables induced by the design matrix $X$, and $V$ captures the covariances $\hat{S}_{\cdot j \cdot l}$ between the response of

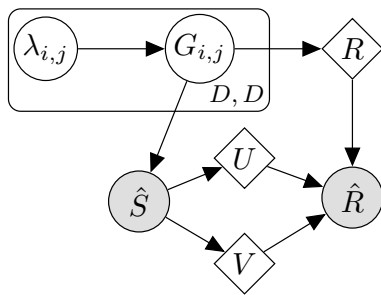

Figure 1: The IBCD model. We treat $\hat{R}$ and $\hat{S}$ as observed and model $\hat{R}$ as coming from a matrix normal distribution $\mathcal{MN}(R, U, V)$ with $U$ and $V$ determined from $\hat{S}$. We place a spike–and–slab horseshoe prior on $G$ which determines the mean matrix $R$ via equation 5.

variables to interventions. This reduces the computational burden of estimating $D^4$ entries in the covariance matrix $\hat{S}$ to estimating the $2D^2$ entries in the covariance matrices $U$ and $V$.

**Estimating the matrix normal covariance matrices.** For linear IV estimators, the entries of the variance-covariance matrix $\hat{S}$ are given by,

$$\hat{S}_{ijkl} = \frac{\text{Cov}(\hat{y}_i, \hat{y}_k)\text{Cov}(\hat{\epsilon}_{ij}, \hat{\epsilon}_{kl})}{n\text{Var}(\hat{y}_i)\text{Var}(\hat{y}_k)}, \tag{9}$$

where $n$ is the total number of observations, $\hat{y}_i$ is the projection of $y_i$ onto the space spanned by its instruments, and $\hat{\epsilon}_{ij} = Y_j - \hat{R}_{ij}Y_i$ is an estimate of the residual (Appendix A). Notice a natural separation into two terms, the first representing covariation of the projections onto the instrument sets $X_i$ and $X_k$, and the second representing covariation in the response of $y_j$ and $y_l$ to upstream interventions. While this response is dependent on the intervened variables, we average over interventions to approximate the the matrix-normal column covariance. As such we can write,

$$\hat{U}_{ik} = \frac{\text{Cov}(\hat{y}_i, \hat{y}_k)}{n\text{Var}(\hat{y}_i)\text{Var}(\hat{y}_k)}, \tag{10}$$

$$\hat{V}_{jl} = \frac{1}{\sum_{ik} \mathbb{1}\{U_{ik} \neq 0\}} \sum_{i,k \mid U_{ik} \neq 0} \text{Cov}(\hat{\epsilon}_{ij}, \hat{\epsilon}_{kl}), \tag{11}$$

where $\mathbb{1}\{U_{ik} \neq 0\}$ indicates non-zero entries in $U$ which are known through the intervention design matrix $X$.

**Overview of the empirical Bayes approach.** We specify an empirical Bayes prior to enforce sparsity over $G$ informed by the structure of the data, $p(G \mid \hat{S})$. Our strategy involves three steps: (1) estimating global inclusion probabilities $\pi$ that reflect the expected graph density under the ER or SF graphs, (2) localizing these probabilities by adjusting each $\pi_{0,ij}$ using correlations in the observational control data, and (3) using these refined edge-specific probabilities to parameterize a spike-and-slab horseshoe prior on the individual edge weights $G_{ij}$.

**Learning ER global prior.** To capture ER structure, we estimate marginal edge inclusion probabilities with a mixture Gaussian to the observed effects $\hat{R}$ (Stephens, 2017), assuming each direct effect $G_{ij}$ follows a unimodal distribution centered at zero. This flexible empirical Bayes prior captures both sparsity and heterogeneous effect sizes while remaining computationally stable compared to traditional spike and slab formulations. Formally, the prior is

$$p\big(G_{ij} \mid \{\pi_k\}, \{\sigma_k^2\}, \hat{s}_{ij}\big) \propto \pi_0 \mathcal{N}\big(0, \sigma_0^2 + \hat{s}_{ij}^2\big) + \sum_{k=1}^{K} \pi_k \mathcal{N}\big(0, \sigma_k^2 + \hat{s}_{ij}^2\big), \tag{12}$$

where $\pi_0$ corresponds to a near-zero spike component and the slab weights $\{\pi_k\}$ are learned over a grid of variances $\{\sigma_k^2\}$. The standard error of the estimate of $\hat{R}_{ij}$, given by $\hat{s}_{ij} = \hat{S}_{ijij}$, accounts for edge specific heteroskedasticity. We fit the parameters $\{\pi_k\}$ to the observed $\hat{R}$ using Expectation Maximization (EM), see Appendix B for details.

**Learning SF global prior.** To capture SF structure, we derive edge inclusion probabilities from the squared total effect matrix $A = |\hat{R}|^2$, which amplifies strong signals and suppresses noise. We compute empirical out-weights $\theta_i = \sum_{j\neq i} A_{ij}$ and in-weights $\phi_j = \sum_{i\neq j} A_{ij}$, and solve the constrained optimization

$$\min_{P_{ij}\in[0,1],\, i\neq j} \sum_{i=1}^{D}\left(\sum_{j\neq i} P_{ij} - \theta_i\right)^2 + \sum_{j=1}^{D}\left(\sum_{i\neq j} P_{ij} - \phi_j\right)^2 \tag{13}$$

subject to $P_{ii} = 0$, to recover a probabilistic edge inclusion matrix $P \in [0,1]^{D\times D}$ that represents the prior probability of an edge existing, $i \to j$. We then transform the solution into spike weights $\pi_{0,ij} = 1 - P_{ij}$.

**Localizing the global prior.** As two variables with a causal effect between them cannot be statistically independent (Vonk et al., 2023), we capture the intuition that more correlated variables are more likely to reflect a causal effect *a priori* by building an edge specific prior, $p(G_{ij}) \propto \mathrm{Cov}(y_i, y_j)$ in the observational (non-intervened) data. Together, this hybrid empirical Bayes strategy leverages both global patterns and local signal to construct a structured, data-driven prior over $G$ that improves both precision and interpretability in causal network inference. To localize the global prior, we calculate the covariance $\Sigma = \mathrm{Cov}(Y_C)$ where $Y_C$ are the non-intervened control samples. We then normalize this into a matrix of interaction strengths $\xi := \frac{\Sigma \odot \Sigma}{||\Sigma \odot \Sigma||_F^2}$ such that each $\xi_{ij} \in [0,1]$. We use this to construct edge specific weights $\pi_k^{ij}$ by solving the convex quadratic program,

$$\min_{\pi_0,\pi_k} \sum_{i\neq j}\left[\left(\pi_k^{i,j} - \xi_{ij}\right)^2 + \left(\pi_0^{i,j} - (1-\xi_{ij})\right)^2\right], \tag{14}$$

subject to the constraints,

$$\pi_0^{i,j} + \pi_k^{i,j} = 1 \quad \forall i \neq j, \tag{15}$$

$$\pi_0^{i,i} = 1, \quad \pi_k^{i,i} = 0 \quad \forall i, \tag{16}$$

$$\pi_0^{i,j} \geq 0, \quad \pi_k^{i,j} \geq 0 \quad \forall i,j, \tag{17}$$

with the sparsity constraint depending on the prior:

$$\text{ER:} \sum_{i\neq j}\pi_0^{i,j} = \pi_0^{\mathrm{global}} \cdot (D^2 - D), \tag{18}$$

$$\text{SF:} \sum_{j>i}\pi_{0,ij} = \pi_{0,i}(D - i - 1) \quad \forall i, \tag{19}$$

where $\pi_0^{\mathrm{global}}$ is the average sparsity level estimated from the EM step from ER, and $\pi_{0,i} \in [0,1]$ reflects the empirical sparsity of node $i$ from SF. We solve this convex optimization problem using the `CVXPY` package (Diamond & Boyd, 2016).

**Spike-and-slab horseshoe prior.** After learning edge specific spike and slab weights separately from the ER and SF priors, we place a prior on each off-diagonal edge $G_{ij}$ in the form of a non-centered spike-and-slab horseshoe:

$$G_{ij} \sim \pi_{0,ij}\mathcal{N}(0,\sigma_0^2) + (1-\pi_{0,ij})(\tau\,\lambda_{ij}\,\epsilon_{ij}), \tag{20}$$

where $\sigma_0^2 = 10^{-3}$, $\epsilon_{ij} \sim \mathcal{N}(0,1)$, $\lambda_{ij} \sim \mathrm{HalfCauchy}(1)$, and $\tau = 0.1$. We use the non-centered reparameterization (Papaspiliopoulos et al., 2007) $\theta_{ij} = \tau\,\lambda_{ij}\,\epsilon_{ij}$ of the hierarchical horseshoe $\theta_{ij} \sim \mathcal{N}(0, \tau^2\lambda_{ij}^2)$, where $\tau$ is a global shrinkage parameter and $\lambda_{ij}$ provides local adaptivity. This reparameterization decouples latent variables from their scales and mitigates Neal's funnel (Neal, 2003) for faster and more stable convergence.

**Sampling from the posterior.** Together, these components define the posterior,

$$p\left(G \mid \hat{R}, \hat{S}\right) \propto p\left(\hat{R} \mid G, \hat{S}\right) p\left(G \mid \hat{S}\right). \tag{21}$$

Our non-centered horseshoe prior produces a geometry that is well-suited for gradient-based inference. We sample from the posterior $p\left(G \mid \hat{R}, \hat{S}\right)$ using the No-U-Turn Sampler (NUTS) (Hoffman et al., 2014), implemented in NumPyro (Bingham et al., 2019; Phan et al., 2019), see Appendix C for details.

## 4 Experiments

We performed comprehensive evaluations of our method as compared to others on both simulated and real data. For simulation, we generated datasets, with a known intervention design matrix, from the model equation 6 using ER and SF DAGs varying dimensions, $D \in \{50, 150, 250, 500\}$. Each dataset included $N = 100$ intervention samples per variable and $D \times 100$ control samples, yielding $2D \times 100$ total observations. This choice of $N = 100$ conservatively matches the average of 141 cells per gene in the Perturb-seq experiment Replogle et al. (2022). In addition, for $D = 50$ we conducted experiments with varying intervention sample sizes, $N \in \{5, 15, 25, 50, 75, 100\}$, under the same design. For real data we analyzed normalized single cell Perturb seq expression matrices from K562 spanning the genome wide and essential gene screens. We retained 521 genes well-captured in both studies for downstream analysis after regressing out S and G2M cell cycle effects, see Appendix D for details. We compare the posterior mean weighted adjacency matrix inferred by IBCD, obtained by averaging marginal edge weights across posterior samples, against various methods including NOTEARS (Zheng et al., 2018), SDCD (Nazaret et al., 2023), BayesDAG (Annadani et al., 2023), LiNGAM (Shimizu et al., 2006), GIES (Hauser & Bühlmann, 2012), GES (Chickering, 2002), inspre (Brown et al., 2025), LLCB (Weinstock et al., 2024), igsp (Yang et al., 2018), and AVICI (Lorch et al., 2022). We threshold small entries in all resultant weighted adjacency matrices to 0 and binarize edge presence. This removes edge weights and signs while preserving directionality. Graph metrics are reported on the resulting directed binary graphs. For purely observational methods (GES, NOTEARS, LiNGAM, BayesDAG) we doubled the number of control samples to match the total sample size of interventional datasets. We report results for CV-tuned sparsity parameter settings of GES and GIES. We note that we have included the popular and easy to run LiNGAM as a reference baseline, even though our simulated data use Gaussian noise.

### 4.1 Scaling with dimension on ER and SF graphs

We evaluated each method by comparing the inferred graph to the ground truth using F1, structural Hamming distance (SHD), AUPRC, and structural intervention distance (SID); AUPRC is reported on all methods for completeness, despite its limited interpretability for methods that produce binary graph estimates. We compared all methods on synthetic datasets with known ground truth graphs averaged over ten replicates from independently generated graphs. Figure 2 summarizes F1, SHD, AUPRC, and SID across dimensions $D \in \{50, 150, 250, 500\}$ when $N = 100 \times D$ intervention and control samples on ER and SF graphs. All optimization and score-based methods scaled to $D = 500$ except NOTEARS which ran to $D = 250$, while Bayesian methods LLCB and BayesDAG exceeded practical time or memory limits beyond $D = 50$; additionally, AVICI did not scale to the $D = 50$ setting. SID is not reported for LLCB due to frequent cyclic outputs, as it is defined only for DAGs. See Table 1 for details.

IBCD achieves the best overall performance, with the highest average F1, AUPRC, and the lowest average SHD, SID in both ER and SF graphs. These results highlight the benefit of a prior that adapts to the underlying graph topology. To ensure these results are not driven by exploitable artifacts (Reisach et al., 2021; 2023), we mitigate varsortability by normalizing all variables to unit variance. Our simulation procedure avoids $R^2$-sortability in ER graphs by design, however the relationship between $R^2$-sortability and SF graphs is nuanced. See Appendix F for a detailed discussion. Detailed per-$D$ tables for F1, SHD, AUPRC, and SID and figures for precision and recall are provided in Tables 8, 9, 10, and 11 and Figure 6.

We observe that NOTEARS and inspre perform better on ER graphs, consistent with theory that says $\ell_1$-regularized methods need more samples to recover hub-heavy SF topologies (Liu & Ihler, 2011). By contrast,

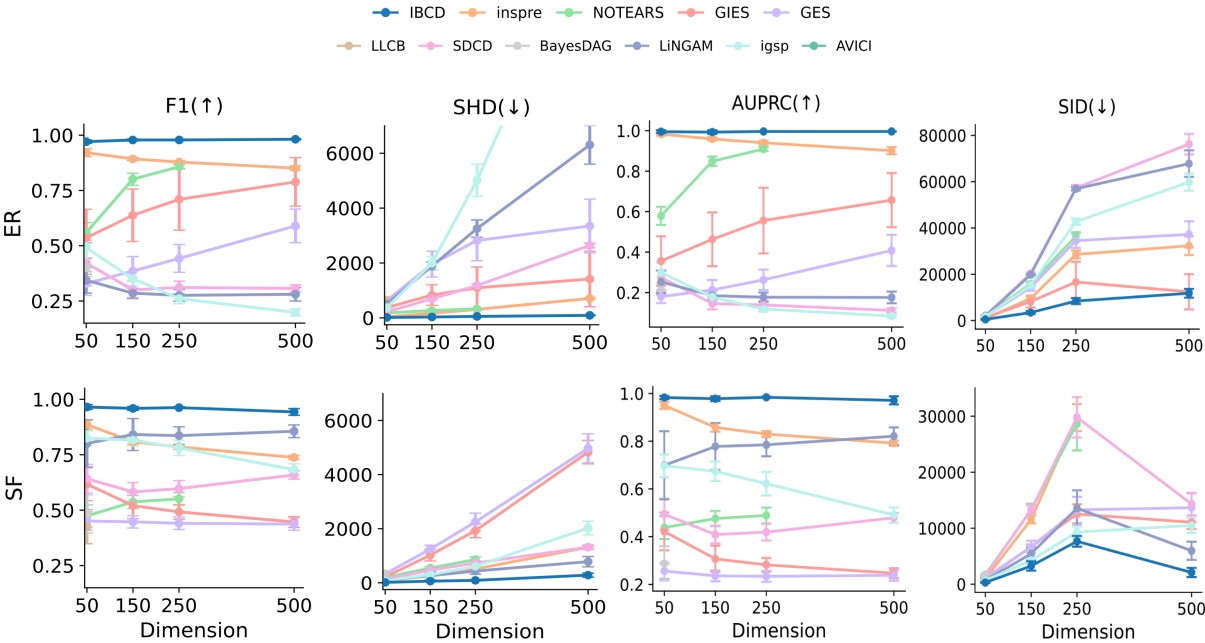

Figure 2: Comparison of F1, SHD, AUPRC, and SID with increasing numbers of dimensions on ER and SF graphs, using 100 intervention samples per dimension.

LiNGAM and SDCD are stronger on SF graphs. LiNGAM first estimates an ordering via independence tests, which we suspect may be easier when a few hubs dominate, and SDCD's spectral-radius penalty seems numerically more stable on SF structures. GIES, the interventional extension of GES, outperforms GES, highlighting the benefit of using interventional data, though on SF graphs the gain can vanish as $D$ increases. We suspect that GES-family methods tend to do better on ER than SF graphs because ER graphs have a more even degree distribution, so the score may change more smoothly with edge edits, yielding fewer near-ties and more reliable greedy steps. In contrast, SF hubs may create many colliders and dense neighborhoods, which yield many near equivalent choices and makes greedy search more prone to suboptimal moves. More formal analysis would be required to validate this intuition.

### 4.2 Scaling with sample size at $D = 50$

We again assess performance as a function of the number of intervention samples per variable $N$ at $D = 50$, reporting F1, SHD, SID, and AUPRC (Figure 3) on both ER and SF graphs. As $N$ increases, most methods gain in F1, AUPRC and drop in SHD, SID with diminishing returns beyond about $N = 50$. IBCD achieves the best overall performance on ER and SF graphs, achieving the highest mean F1, AUPRC and lowest SHD, SID in almost all settings, except the low-sample case ($N = 5$). Detailed per-$N$ tables for F1, SHD, AUPRC, and SID and figures for precision and recall are provided in Tables 12, 13, 14, and 15 and Figure 7.

### 4.3 Expected graph recovery performance

Our method yields full MCMC posterior samples of the causal graph, allowing computation of metrics such as expected SHD (E-SHD) and F1 (E-F1). Throughout the paper, we emphasize single-graph metrics, since our primary goal is to provide a graph estimate together with edge-level uncertainty, rather than to fully characterize the posterior. In contrast, most baseline methods return only a single graph estimate, making E-metrics infeasible without computationally expensive bootstrap procedures. To highlight the practical value of edge-level uncertainty, we analyze posterior edge confidence on large synthetic graphs (Section 4.5) and real Perturb-seq data (Section 4.6). Nevertheless, for completeness, we additionally report E-SHD and E-F1

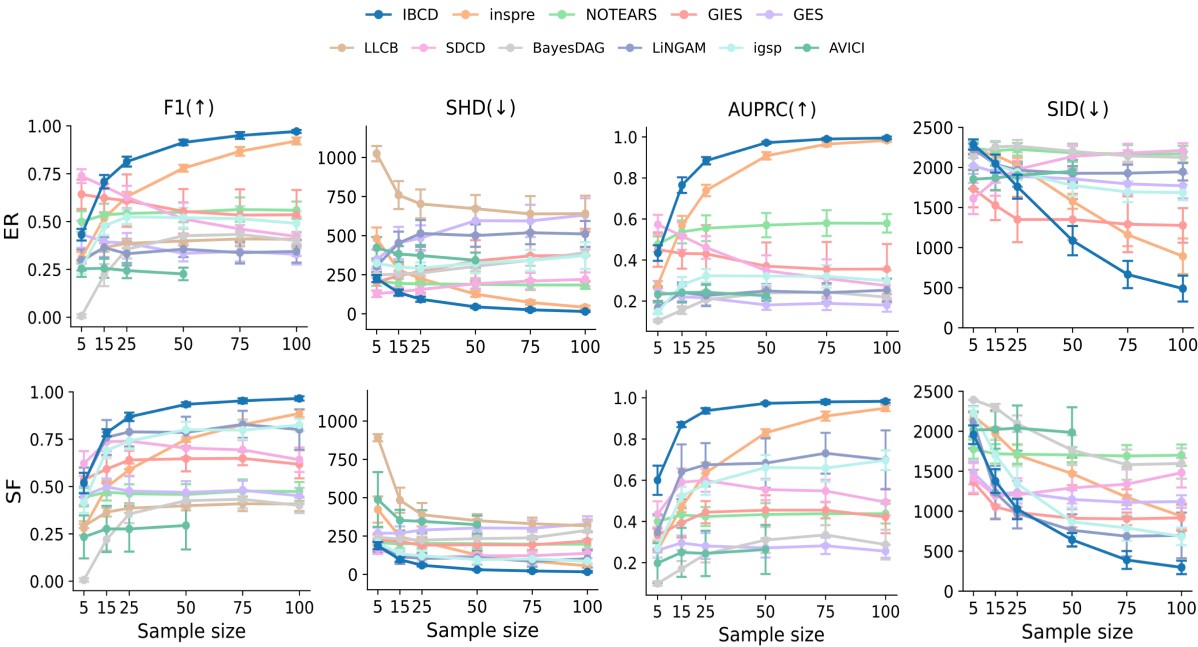

Figure 3: Comparison of F1, SHD, AUPRC, and SID with increasing numbers of intervention sample sizes on ER and SF graphs with $D = 50$.

| Dim | Sample | IBCD | inspre | GIES | GES | LiNGAM | SDCD | NOTEARS | BayesDAG | LLCB | igsp | AVICI |
|---|---|---|---|---|---|---|---|---|---|---|---|---|
| 50D | 5 | 2m 59s | 16s | 1.3s | 0.6s | 1.4s | 30s | 37s | 1m 52s | 10h 17m | 0.2s | 0.4s |
| 50D | 15 | 3m 08s | 15s | 0.8s | 0.9s | 2.6s | 1m 17s | 53s | 2m 49s | 10h 37m | 0.2s | 0.6s |
| 50D | 25 | 3m 47s | 15.7s | 0.9s | 1.1s | 3.8s | 2m 06s | 1m 14s | 3m 37s | 10h 41m | 0.2s | 0.8s |
| 50D | 50 | 4m 11s | 14.9s | 2.1s | 1.4s | 6.9s | 3m 49s | 2m 06s | 4m 58s | 11h 24m | 0.2s | 1.8s |
| 50D | 75 | 4m 58s | 14.8s | 2.5s | 4.1s | 10.8s | 5m 17s | 3m 18s | 5m 59s | 10h 13m | 0.2s | NA |
| 50D | 100 | 6m 20s | 15.4s | 2.0s | 1.9s | 14.2s | 6m 46s | 3m 34s | 7m 07s | 10h 39m | 0.3s | NA |
| 150D | 100 | 14m 30s | 4m 41s | 37s | 46s | 5m 03s | 26m 09s | 1h 10m | NA | NA | 1.6s | NA |
| 250D | 100 | 24m 27s | 19m 41s | 2m 25s | 3m 28s | 22m 02s | 50m 05s | 2h 30m | NA | NA | 5.1s | NA |
| 500D | 100 | 45m 38s | 2h 21m | 20m 15s | 41m 59s | 2h 55m | 1h 43m | NA | NA | NA | 30.7s | NA |

Table 1: Runtimes averaged across 10 replicates per setting, aggregated over ER and SF. All methods ran on a shared compute cluster with Intel Xeon E5-2660 v2 CPUs. IBCD and SDCD used NVIDIA H100 GPUs (80 GB VRAM); IBCD leveraged three GPUs to parallelize MCMC chains. NA indicates failed or unsupported runs (12 hour timeout or resource limits).

for Bayesian baselines with posterior samples and for frequentist methods when 1,000 bootstrap resamples were computationally feasible ($D = 50$, $N = 100$), where our approach consistently outperforms competing methods (Table 7)

### 4.4 Ablation studies

To assess the contribution of each modeling component, we conducted ablation studies comparing different configurations on $D$=50 synthetic ER and SF graphs in Table 2. The ER baseline learns an empirical prior from $\hat{R}$ with a matrix normal likelihood and edge specific weights. An oracle prior that uses the true graph gives only a small, statistically insignificant gain, showing that the observational summary $\hat{R}$ provides a strong empirical prior (see Appendix E). To validate the likelihood we replaced the matrix normal with a multivariate normal that uses a truncated singular value decomposition of the covariance $S$ keeping the top $r = 10$ components which reduces cost from $\mathcal{O}(D^3)$ to $\mathcal{O}(Dr^2)$ but reduced performance highlighting that learning structured row and column covariance matters. Using a single global prior also hurt which supports the value of edge specific priors obtained by localizing the global prior, highlighting the value of adaptive

| Variant | Empirical prior | MN | Edge specific | MVN | Global Prior | ER prior | SF prior | F1 | SHD |
|---|---|---|---|---|---|---|---|---|---|
| ER baseline | ✓ | ✓ | ✓ | × | × | ✓ | × | $0.875 \pm 0.023$ | $58.9 \pm 10.10$ |
| Oracle Prior | × | ✓ | ✓ | × | × | ✓ | × | $0.876 \pm 0.017$ | $59.9 \pm 9.02$ |
| MVN Likelihood | ✓ | × | ✓ | ✓ | × | ✓ | × | $0.431 \pm 0.017$ | $625.4 \pm 18.75$ |
| Global Prior | ✓ | ✓ | × | × | ✓ | ✓ | × | $0.845 \pm 0.016$ | $72.6 \pm 6.06$ |
| ER with SF Prior | ✓ | ✓ | ✓ | × | × | × | ✓ | $0.908 \pm 0.017$ | $44.5 \pm 8.34$ |
| SF baseline | ✓ | ✓ | ✓ | × | × | × | ✓ | $0.894 \pm 0.020$ | $47.6 \pm 8.44$ |
| SF with ER Prior | ✓ | ✓ | ✓ | × | × | ✓ | × | $0.809 \pm 0.025$ | $83.2 \pm 11.42$ |

Table 2: Ablation study results comparing mean and standard deviation of F1 and SHD metrics across model variants averaged over 10 replicates of 50D ER graph. ✓ indicates the presence of each component. MN indicates matrix normal, MVN indicates multivariate normal, ER indicates Erdős–Rényi, and SF indicates Scale-Free.

| Graph | Metric | 50D | 150D | 250D | 500D |
|---|---|---|---|---|---|
| ER baseline | F1 | $0.875\pm0.023$ | $0.848\pm0.019$ | $0.858\pm0.011$ | $0.877\pm0.015$ |
| | SHD | $58.9\pm8.34$ | $216.8\pm25.36$ | $335.9\pm25.86$ | $560.0\pm56.47$ |
| ER + SF prior | F1 | $0.908\pm0.017$ | $0.836\pm0.022$ | $0.789\pm0.035$ | $0.576\pm0.131$ |
| | SHD | $44.5\pm8.34$ | $255.8\pm33.19$ | $598.0\pm120.70$ | $3160.8\pm1502.28$ |

Table 3: Ablation study comparing the mean and standard deviation of F1 and SHD metrics between the ER baseline and ER with SF prior across different graph dimensions, averaged over ten replicates.

sparsity. Surprisingly, using an SF prior on ER graphs improves performance at $D = 50$ but degrades as $D$ grows, with large drops in F1 and increases in SHD (Table 3). In contrast, the SF baseline outperforms SF with an ER prior, indicating that when the topology is unknown the SF baseline is a safer default.

### 4.5 Posterior edge confidence on 500 nodes graphs

To verify that our $D = 500$ inference is not only scalable but also trustworthy at the edge level, we analyzed PIP equation 3, with $\epsilon = 0.05$. The PIP quantifies posterior support for edge presence. If PIPs are appropriately calibrated, approximately 80% of edges with PIP = 0.80 represent true direct causal effects. Figure 4 shows that our reported PIPs are conservative for both ER and SF graphs in high-dimensional settings. Thus, PIPs can be used to assess confidence in individual edges even in difficult inference regimes.

### 4.6 Analysis of real Perturb-seq data

Finally, we assess the performance of IBCD on real data. As there is a scarcity of large-scale data with ground truth causal graphs, we used five-fold stratified cross validation on 521 genes in the essential and genome-wide Perturb-seq screens (Replogle et al., 2022). For each fold $f \in \{1, \ldots, 5\}$, we fit the model on the training cells and computed $\text{PIP}^{(f)}$ on the held-out cells. The scatter plot in Figures 5a and 5b overlays all ten unordered fold pairs $\left(\text{PIP}^{(i)}, \text{PIP}^{(j)}\right)$ $(i < j)$ for the essential and GWPS screens, respectively. Each point is one putative edge, axes are on a log scale, and the top panel shows the marginal distribution of PIP values. Both figures show that the global Pearson correlation across the ten fold pairs is higher with the SF prior than with the ER prior. This is consistent with hub-dominated, heterogeneous gene regulatory networks, so a SF prior better matches the data. Adapting the prior to the topology improves the reproducibility of edge-level PIPs across sub-samples and supports the reliability of our 521-gene inference.

As the genome-wide and essential screens represent independent experiments with inevitable differences in unmeasured exogenous factors, we conduced an out of distribution validation comparing the edge PIPs in both datasets. We fit IBCD on each full dataset and obtained PIPs for all edges. We then compared PIP from the genome wide screen with PIP from the essential screen and observed a strong positive correlation ($r = 0.570$ in SF prior, Figure 8). Again the correlation is much higher with an SF prior than with an ER prior which is consistent with hub dominated gene regulation in these cells.

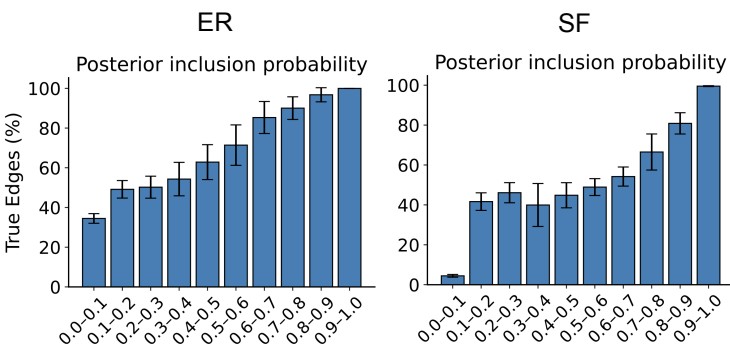

Figure 4: Calibration of posterior inclusion probability on 500D graphs under ER (left) and SF (right) graph in Figure 2. Bars show mean ± s.d. over ten replicates. True edges concentrate in high PIP, confirming reliable uncertainty quantification.

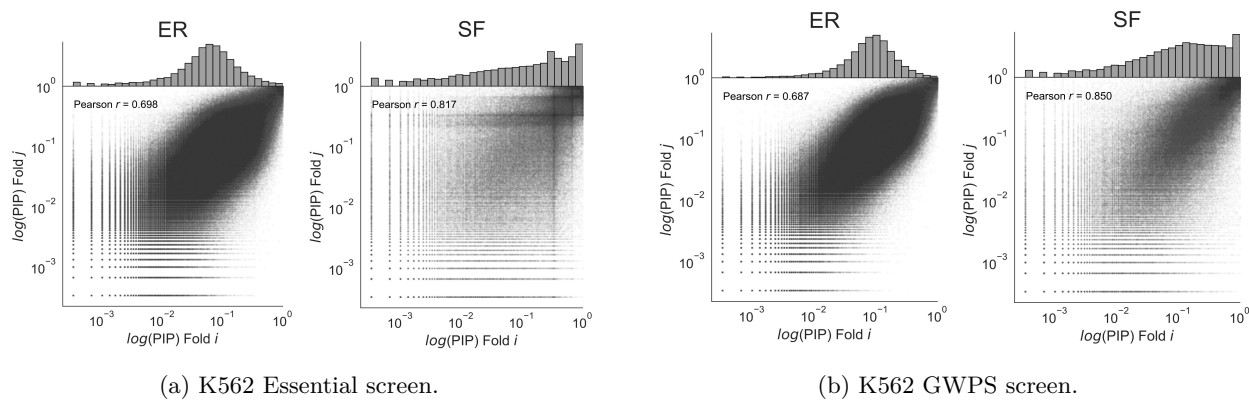

(a) K562 Essential screen.  (b) K562 GWPS screen.

Figure 5: Agreement of log-transformed PIP across the ten cross-validation fold pairs for the essential and GWPS screens. Each point is one edge; the density of points forms a grey cloud. The positive correlation shows that high-PIP edges are consistently identified across folds.

## 5  Discussion

We have introduced Interventional Bayesian Causal Discovery (IBCD), an empirical Bayesian framework that scales Bayesian causal inference to hundreds of nodes while providing calibrated edge level uncertainty. IBCD departs from most existing work in two key ways. First, we treat experimentally perturbed variables as instruments using a soft intervention framework and summarize the data by the matrix of total causal effects, allowing a matrix normal likelihood that reduces the computational burden of likelihood evaluation by orders of magnitude. Second, we learn a spike-and-slab horseshoe prior, with data-driven global sparsity and edge specific weights adapted to the local covariance structure in the observational samples. We also fit topology aware priors separately for ER and SF regimes, allowing the learned prior to match the graph family while using the same inference engine. Third, we show that PIP estimates are stable across cross validation folds and highly correlated across independent experiments, with stronger correlation under the SF prior. This indicates that IBCD yields reproducible edge probabilities on real interventional data at scale.

However, the present work has several limitations that point toward clear avenues for improvement. First, we make several assumptions. Perhaps the most limiting is the assumption of a linear model. While non-linearities are sure to exist, the linear model is well-studied and represents the classic tradeoff between expressivity and tractability. We also assume that every variable has at least one instrument. While these types of large-scale screens are becoming common in genomics, this limits applicability in scenarios where interventions are more difficult. Second, while we make no assumption on the distribution of error terms, we assume that the IV

estimator used to produce causal effect estimates is asymptotically normal. Relatedly, we focus on 2SLS for IV estimates for simplicity, which assumes independence between the instrument and confounders that might not hold in practice. However, note that this framework allows any IV estimator with an asymptotic normal distribution to be used instead. Finally, our aim is to summarize the posterior via the mean graph and quantify edge level uncertainty through PIPs, rather than to characterize the full posterior. This enables straightforward comparisons with non-Bayesian methods but leaves important questions regarding the modal distribution of the posterior open. Relatedly, we do not explicitly enforce an acylicity constraint. This is by design, as an unbiased estimate may lie outside the constraint set. We explored adding a soft spectral-radius penalty, but found it did not improve performance, and that our standard estimates typically do not have a large spectral radius (Table 6 in Appendix G).

By providing scalable, uncertainty aware causal discovery, IBCD can help prioritize high confidence direct causal effects. This can inform experimental follow-up while flagging those that remain ambiguous, thereby reducing costly false leads or guiding experimental design to clarify ambiguous connections. More broadly, we have demonstrated that the combination of instrumental variable summaries with empirical Bayes shrinkage is a powerful, efficient approach to causal discovery with high-dimensional intervention data.

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

## A    Covariance structure of total causal effect estimates

Our goal is to estimate the covariance between the total causal effect estimates, given by

$$\text{Cov}(\hat{R}_{ij}, \hat{R}_{kl}) = \mathbb{E}[(\hat{R}_{ij} - \mathbb{E}[\hat{R}_{ij}])(\hat{R}_{kl} - \mathbb{E}[\hat{R}_{kl}])] \tag{22}$$

We assume the estimator takes a two-stage linear form, such that the function linking the instruments to the exposure and the exposure to the outcome are both linear:

$$\hat{y}_i = P_i y_i, \tag{23}$$
$$\hat{y}_j = \hat{R}_{ij} y_i, \tag{24}$$
$$\hat{R}_{ij} = \hat{Q}_i y_j, \tag{25}$$

where we have written $P_i$ as the projection matrix onto the instruments for $i$, and $\hat{Q}_i$ as the regression operator that determines the coefficients from $\hat{y}_i$. For example, in generalized least squares we would write $P_i = X_i(X_i^\top W_i X_i) X_i^\top W_i$ and $\hat{Q}_i = (\hat{y}_i^\top W_y \hat{y}_i)^{-1} \hat{y}_i^\top W_y$ for weight matrices $W_i$ and $W_y$. We assume the estimators are unbiased so that $\forall i, j, \; \mathbb{E}[\hat{R}_{ij}] = R_{ij}$.

Using 6, we can write $y_j$ as

$$y_j = X\beta R_{:j} + \epsilon R_{:j}, \tag{26}$$

where $R_{:j}$ refers to the $j$'th column of $R$. Using this, we can write

$$\hat{R}_{ij} = \hat{Q}_i(X\beta R_{:j} + \epsilon R_{:j}). \tag{27}$$

Similarly, writing $y_i$ as $y_i R_{ij} = X\beta R_{:i} R_{ij} + \epsilon R_{:i} R_{ij}$ and adding and subtracting $\hat{Q}_i y_i$ to 27 and collecting terms we have,

$$\hat{R}_{ij} = \hat{Q}_i(X\beta R_{:j} + \epsilon R_{:j}) + \hat{Q}_i y_i R_{ij} - \hat{Q}_i y_i R_{ij} \tag{28}$$
$$= \hat{Q}_i(y_i R_{ij} + X\beta R_{:j} - X\beta R_{:i} R_{ij} + \epsilon R_{:j} - \epsilon R_{:i} R_{ij}) \tag{29}$$
$$= R_{ij} + \hat{Q}_i(X\beta R_{:j} - X\beta R_{:i} R_{ij} + \epsilon R_{:j} - \epsilon R_{:i} R_{ij}). \tag{30}$$

Because $\hat{Q}_i$ projects onto instruments for $y_i$, all effects $X\beta R_{:j}$ must travel through $y_i$, so $\hat{Q}_i X\beta R_{:j} = \hat{Q}_i X\beta R_{:i} R_{ij}$. This leaves us with,

$$\hat{R}_{ij} - R_{ij} = \hat{Q}_i(\epsilon R_{:j} - \epsilon R_{:i} R_{ij}) \tag{31}$$
$$= \hat{Q}_i(y_j - R_{ij} y_i) \tag{32}$$
$$= \hat{Q}_i \epsilon_{ij}. \tag{33}$$

Using this and independence across samples we see that the covariance entries are given by

$$\text{Cov}(\hat{R}_{ij}, \hat{R}_{kl}) = \mathbb{E}[\hat{Q}_i \epsilon_{ij} \epsilon_{kl}^\top \hat{Q}_k^\top] \tag{34}$$
$$= \hat{Q}_i \hat{Q}_k^\top \text{Cov}(\epsilon_{ij}, \epsilon_{kl}). \tag{35}$$

In the least squares case this leads to the plug in estimator 9,

$$\hat{S}_{ijkl} = \frac{\text{Cov}(\hat{y}_i, \hat{y}_k)\text{Cov}(\hat{\epsilon}_{ij}, \hat{\epsilon}_{kl})}{n\text{Var}(\hat{y}_i)\text{Var}(\hat{y}_k)}.$$

## B    Prior estimation

### B.1    ER global sparsity prior estimation

The goal of the global sparsity prior is to estimate a flexible, data-driven mixture prior over all off-diagonal entries of the total effect matrix $\hat{R}$, capturing both sparsity and a range of plausible effect sizes. We require

the following constraints:

$$\pi_0 + \sum_{k=1}^{K} \pi_k = 1, \quad \pi_0 \geq 0, \quad \pi_k \geq 0. \tag{36}$$

To estimate the mixture weights $\{\pi_0, \pi_1, \ldots, \pi_K\}$, we maximize a penalized likelihood over the observed $\{w_i\}_{i=1}^{N}$. The log-likelihood of the data under this model is:

$$\log L(\pi_0, \pi_1, \ldots, \pi_K) = \sum_{i=1}^{N} \log \left[ \pi_0 f_0(w_i) + \sum_{k=1}^{K} \pi_k f_k(w_i) \right], \tag{37}$$

where $f_0(w_i) = \mathcal{N}(w_i; 0, \sigma_0^2 + \hat{s}_i^2)$ and $f_k(w_i) = \mathcal{N}(w_i; 0, \sigma_k^2 + \hat{s}_i^2)$ incorporate edge specific standard errors $\hat{s}_i$ from IV regression to reflect heteroskedasticity across edges. We add a Dirichlet-like penalty to the slab components:

$$\text{Obj}(\pi_0, \pi_1, \ldots, \pi_K) = \sum_{i=1}^{N} \log \left[ \pi_0 f_0(w_i) + \sum_{k=1}^{K} \pi_k f_k(w_i) \right] + (\alpha - 1) \sum_{k=1}^{K} \log \pi_k, \tag{38}$$

where $\alpha = 2$ by default. A larger $\alpha$ encourages use of the slab components. We optimize this penalized objective using the EM algorithm. In the E-step, for each $w_i$, we compute the posterior responsibility for each mixture component:

$$z_{i,0} = \frac{\pi_0 f_0(w_i)}{\pi_0 f_0(w_i) + \sum_{m=1}^{K} \pi_m f_m(w_i)}, \tag{39}$$

$$z_{i,k} = \frac{\pi_k f_k(w_i)}{\pi_0 f_0(w_i) + \sum_{m=1}^{K} \pi_m f_m(w_i)}. \tag{40}$$

We then compute expected counts:

$$n_0 = \sum_{i=1}^{N} z_{i,0}, \quad n_k = \sum_{i=1}^{N} z_{i,k}, \quad k = 1, \ldots, K. \tag{41}$$

In the M-step, we update the mixture weights by maximizing the complete-data penalized log-likelihood:

$$\pi_0 = \frac{n_0}{N + (\alpha - 1)K}, \tag{42}$$

$$\pi_k = \frac{n_k + (\alpha - 1)}{N + (\alpha - 1)K}, \quad k = 1, \ldots, K. \tag{43}$$

These updates ensure that the weights remain normalized and non-negative.

## C   Inference hyperparameters

We used No-U-Turn Sampler (NUTS), with a target acceptance probability of 0.7, which balances step size and acceptance rate in Hamiltonian Monte Carlo, and a maximum tree depth of 10 to ensure thorough posterior exploration without excessive computation. The sampler is initialized at the median of three prior predictive draws, providing a stable starting point for sampling. We ran three chains with 300 warm-up steps and 1000 samples per chain.

## D   Experiment details

### D.1   Synthetic data generation and processing

To evaluate performance across varied causal regimes, we used the simulator from (Brown et al., 2025). We generated synthetic datasets with a known intervention design matrix from the model 6 using both Erdős–Rényi

(ER) and scale-free (SF) graph topologies while varying the graph size, $D \in \{50, 150, 250, 500\}$. To keep sparsity comparable across sizes, we targeted an average node degree of approximately 5. For ER, we set the edge inclusion probability to $p \approx 5/D$, specifically $p = \{0.10, 0.033, 0.02, 0.01\}$ for $D = \{50, 150, 250, 500\}$. For SF, we empirically calibrated $p$ per $D$ to achieve an average degree of 5, selecting $p = \{0.066, 0.108, 0.124, 0.139\}$ for $D = \{50, 150, 250, 500\}$ via repeated simulation. Edge weights were drawn from a PERT distribution (Clark, 1962) with minimum $v/2$, mode $v$, and maximum $2v$, where $v$ controls the typical magnitude of causal effects on a per-variance scale, and we set $v = 0.25$. All data were normalized to mean 0, variance 1 in the control samples. To specify the strength of interventions, we defined the effect size $\beta$ as the number of standard deviations by which the mean of $Y_i$ shifts when the intervention $X_i = 1$ is applied. The intervention strength was fixed at $\beta = -2$. Each configuration included $D \times 100$ interventional samples and $D \times 100$ control samples ($2D \times 100$ total). Additionally, for $D = 50$ we created datasets with varying intervention sample sizes, $N \in \{5, 15, 25, 50, 75, 100\}$, under the same design. For baselines that operate on observational data only (GES, NOTEARS, LiNGAM, BayesDAG), we generated purely observational datasets with $2D \times 100$ control samples ($N = 0$) so that the total sample size matches the interventional datasets. For ablations we set $\beta = -1.5$ and $v = 0.2$ and used the same graph construction with ER graphs at $D \in \{50, 150, 250, 500\}$ and SF graphs only at $D = 50$ with 100 intervention samples per variable and $D \times 100$ control samples. All configurations were replicated across ten replicates to assess robustness of the methods.

### D.2 K562 Perturb-seq Data and processing

For real data, we analyzed both the genome-wide (GWPS) and essential Perturb-seq screens in K562 cells (Replogle et al., 2022). We considered genes present in both panels and retained genes where the estimated per-variance effect-size of the intervention on the target gene was at least $-0.75$ standard deviations, and where there were at least 50 cells receiving a guide targeting that gene. This yielded 521 well-captured genes for downstream analysis. Raw single-cell matrices were obtained from Figshare (`https://plus.figshare.com/ndownloader/files/35773075`) to regress out cell-cycle effects prior to downstream inference. This dataset contains CRISPR-inhibition guide RNA targeting each of 2,057 essential genes in the essential panel, and targeting approximately 9,866 expressed genes in the GWPS panels, with each cell receiving one guide targeting one gene.

To enhance the perturbation signal and reduce cell cycle confounding, we regress out per-cell S and G2M module scores prior to inference. Let $Y \in \mathbb{R}^{C \times G}$ denote the log-normalized expression matrix after size-factor normalization, where $c$ indexes cells and $g$ indexes genes. Let $s_c$ be the S phase module score and $m_c$ the G2M module score, computed per cell from standard S and G2M gene sets with expression matched control genes. For each gene $g$, we fit

$$Y_{cg} = \alpha_g + \beta_g^{(S)} s_c + \beta_g^{(G2M)} m_c + \varepsilon_{cg}.$$

We define residuals using the fitted coefficients, $\hat{\varepsilon}_{cg} = Y_{cg} - \hat{\alpha}_g - \hat{\beta}_g^{(S)} s_c - \hat{\beta}_g^{(G2M)} m_c$. Within each 10x GEM group $h$, we compute $\mu_{gh}^{\text{NTC}}$ and $\sigma_{gh}^{\text{NTC}}$ as the mean and standard deviation of $\hat{\varepsilon}_{cgh}$ over non-targeting control (NTC) cells in group $h$ and standardize residuals per gene using these NTC statistics within each group, $Z_{cgh} = \frac{\hat{\varepsilon}_{cgh} - \mu_{gh}^{\text{NTC}}}{\sigma_{gh}^{\text{NTC}}}$. Here $Z_{cgh}$ denotes the standardized residual for cell $c$, gene $g$, and GEM group $h$. This yields a cell-cycle–adjusted and GEM-normalized matrix $Z$ for downstream analysis. We used 2SLS to estimate causal effects using guide presence as an instrumental variable. Note that because each cell receives one guide, there is no covariance between $\hat{y}_i$ and $\hat{y}_k$ when $i \neq k$. Thus, the covariance 9 simplifies to

$$\hat{S}_{ijkl} = \begin{cases} \frac{1}{n\hat{\beta}_i^2} \text{Cov}(\hat{\epsilon}_{ij}, \hat{\epsilon}_{il}) & \text{if } i = k, \\ 0 & \text{otherwise.} \end{cases}$$

For each gene, we estimated the effect size $\hat{\beta}_i$ of the instrument on its target gene using linear regression of gene expression on perturbation status. We then restricted analyses to the 521 genes shared by the GWPS and essential panels.

### D.3 Baselines

To evaluate our method in a wider context, we compare our model (IBCD) to a diverse set of different methods We describe our baselines briefly in the following.

- **inspre (Brown et al., 2025)** inverse sparse regression (inspre) learns a sparse left inverse of the average causal effect matrix from per target interventions and reconstructs a weighted causal network. We use the implementation at `https://github.com/brielin/inspre` with default hyperparameters and set `DAG=TRUE` to enforce acyclicity.

- **LiNGAM (Shimizu et al., 2006)** LiNGAM assumes linear relations, acyclicity, and independent non-Gaussian noise to recover a causal ordering and estimates edge weights by regression. We used the implementations from the `pcalg` R package with default parameter settings.

- **GES (Chickering, 2002)** Greedy Equivalence Search (GES) is a greedy score-based method for causal discovery using the BIC score. We used the implementations from the `pcalg` R package, where the default sparsity penalty $\lambda = 0.5$ is used unless otherwise specified; tuned sparsity values for GES is reported in Table **??**.

- **GIES (Hauser & Bühlmann, 2012)** Greedy Interventional Equivalence Search (GIES) is an interventional extension of GES (Chickering, 2002) that uses known intervention targets to score structures and orient edges. We used the implementations from the `pcalg` R package, where the default sparsity penalty $\lambda = 0.5$ is used unless otherwise specified; tuned sparsity values for GIES is reported in Table **??**. We include GIES to highlight the benefit of intervention data, and in our experiments it recovered graph structure better than the observational-only GES.

- **SDCD (Nazaret et al., 2023)** Stable Differentiable Causal Discovery (SDCD) is a differentiable, score based method that enforces acyclicity with a spectral radius penalty and trains in two stages, edge preselection then penalized differentiable causal discovery, to improve stability and scale. We use the implementation at `https://github.com/azizilab/sdcd` with default parameters, setting `thresholding=True` to enable the built-in, data-adaptive thresholding driven by inferred effect sizes, following the jupyter notebook tutorial.

- **NOTEARS (Zheng et al., 2018)** NOTEARS reformulates score-based DAG learning as a smooth equality-constrained optimization using a matrix-exponential acyclicity function, enabling global continuous search with standard solvers. We use the implementation at `https://github.com/xunzheng/notears` and set `w_threshold`=0 to apply thresholding uniformly in post-processing. Our simulations use unit-normalized data such that edge weights represent per-variance effect sizes; therefore the default threshold 0.3 is well above the median simulated effect size and would degrade performance.

- **BayesDAG (Annadani et al., 2023)** BayesDAG combines hybrid strategy where MCMC learns permutations and mechanism parameters, while variational inference infers DAG edges given the learned permutations. We use the implementation at `https://github.com/microsoft/Project-BayesDAG` with default settings.

- **LLCB (Weinstock et al., 2024)** Linear Latent Causal Bayes (LLCB) is a Bayesian extension of Linear Latent Causal model (Hyttinen et al., 2012) that deconvolves total perturbation effects into direct gene–gene edges using sparsity and spectral-radius priors using a variational method, Pathfinder (Zhang et al., 2022). We use the implementation at `https://github.com/weinstockj/LLCB` with default settings.

- **igsp (Yang et al., 2018)** Interventional greedy sparsest permutation (igsp) is a permutation-based method that recovers the equivalence classes of graph. We use the implementation at `https://github.com/asxue/dotears/blob/main/workflow/scripts/igsp.py`, with cutoffs $\alpha = 0.001$ for conditional independence tests and for invariance tests.

- **AVICI (Lorch et al., 2022)** Amortized variational inference for causal discovery (AVICI) reframes causal structure learning as an amortized inference problem using a neural network that directly predicts causal graphs from observational or interventional samples. We use the implementation at `https://github.com/larslorch/avici` with the default pretrained `scm-v0` model.

## E   Oracle priors from the true graph

In our ablation studies the oracle prior uses the true matrix $G$ to directly set the spike probability and serves as an upper bound that isolates the impact of learning priors from observational summaries. We flatten the off diagonal entries of $G$ into $w = (w_1, \ldots, w_N)$ with $N = D(D-1)$. We compute

$$\pi_0 = \frac{1}{N} \sum_{i=1}^{N} \mathbb{1}\{w_i = 0\}.$$

The slab mass is $1 - \pi_0$. These weights parameterize the spike and slab horseshoe prior over $G$ used in the ablation studies.

## F   On $R^2$-sortability, SF graphs and SID

### F.1   Default variance-balanced simulations and emergent $R^2$-sortability

Causal discovery benchmarks can sometimes be misleading if the simulated data contains structural artifacts that allow algorithms to guess the causal ordering without actually learning the underlying data (Reisach et al., 2021; 2023). Two such artifacts are varsortability, where the marginal variance of nodes increases along the causal order, and $R^2$-sortability, where the proportion of variance explained by a node's parents is correlated with its position in the graph. To ensure our results are driven by causal signal rather than these artifacts, we simulate nodes to have a uniform unit variance by balancing the noise contribution in  D.1.

1. We generate an ER or SF random DAG $G$ with edge weights drawn i.i.d. from a specified PERT distribution (Clark, 1962) with mode 0.3.

2. For each node, we compute the variance explained by its causal parents as the sum of squared incoming edge weights, and set the noise variance to $1 - \sum_j G_{ij}^2$, ensuring that the expected total variance of each node is one.

3. Finally, we simulate data from the standard linear autoregressive model and normalize each variable to have exactly unit variance after sampling.

We reasoned that this variance-balancing procedure would mitigate both varsortability and $R^2$-sortability by preventing the relative contribution of noise variance from vanishing as the causal order increases, which is a primary driver of these artifacts. Empirically, however, this construction exhibits qualitatively different behavior in ER and SF graphs. In ER graphs, this simulation scheme produces data that are not $R^2$-sortable on average. Across 100 randomly generated ER graphs with average degree 100 and 5,000 observational samples, the mean $R^2$-sortability statistic is 0.52±0.09, indicating little correlation between the fraction of variance explained by a node's parents and its position in the causal order. In contrast, when applied to SF graphs, the same procedure yields anti-$R^2$-sortable data where for over 100 randomly generated SF graphs, the mean $R^2$-sortability statistic is 0.05±0.02, indicating that later nodes tend to have less variance explained by their parents.

We find that this phenomenon arises from structural properties of SF graphs. As node degree decreases along the causal order in SF graphs, enforcing unit marginal variance requires injecting progressively larger noise variance for earlier nodes. As a result, the amount of added noise becomes positively correlated with causal order, inducing a negative correlation between explained variance and position in the graph. We note that despite this effect, the relative performance of IBCD remains strong in both ER and SF settings. Moreover,

prior work has highlighted that avoiding $R^2$-sortability artifacts in scale-free graphs is particularly challenging (Reisach et al., 2023), suggesting that this issue is not unique to our simulation design.

Avoiding anti-$R^2$-sortability in this setting would require departing from i.i.d. edge-weight sampling. In particular, the expected magnitude of individual causal effects would need to increase with causal order, or equivalently decrease with the number of parents, such that the total variance explained remains constant across nodes despite degree decay. While such constructions are possible, they impose additional structure on how causal effect sizes depend on graph position.

### F.2 Explicit control of $R^2$-sortability in SF graphs

To explore alternative simulation designs, we additionally consider schemes that explicitly control $R^2$-sortability in SF graphs. In a linear structural equation model with weighted adjacency matrix $G$, the coefficient of determination for regressing node $i$ on all others is given by

$$R_i^2 = C_i^\top C_{-i}^{-1} C_i,$$

where $C = (I - G)^{-1\top}(I - G)^{-1}$ is the data covariance, $C_{-i}$ is marginalization over $i$, dropping the $i$th row and column, and $C_i$ is the $i$th row of C with entry $i$ removed. A graph $G$ therefore has an $R^2$-sortability metric of 0.5 if and only if $C_i^\top C_{-i}^{-1\top} C_i \perp i$.

Given an arbitrary G, one can therefore produce a new graph $G' = G \circ A$ that is not $R^2$-sortable by finding a matrix $A$ and constant $c$ such that $\forall_i C_i'^\top C_{-i}'^{-1\top} C_i' = c$, with $C' = (I - G \circ A)^{-1\top}(I - G \circ A)^{-1}$. Rather than solving this condition directly, we adopt a practical approximation motivated by the interpretation of edge weights in our simulations. The entries of $G$ in our simulation represent per-variance causal effects, with the variance in $Y_j$ explained by $Y_i$ given by $G_{ij}^2$. We can then compute the total incoming and outgoing strength of each node as

$$a = \sum_i G_{ij}^2 + \sum_j G_{ij}^2$$

and approximate the weight matrix as the outer product $A = \frac{1}{\sqrt{a}}\frac{1}{\sqrt{a}}^\top$. We then form $G' = G \circ A$ and renormalize $G'$ such that the mean edge weight in $G$ and $G'$ are equal.

This reweighting scheme substantially reduces $R^2$-sortability in practice. On $D = 250$ SF graphs, the mean $R^2$-sortability statistic across 10 random graphs is $0.32 \pm 0.05$, indicating weak $R^2$-sortability. However, this have substantial structural consequences on $G$. Specifically, since scale-free graphs exhibit an exponentially decaying degree distribution, requiring nodes early in the causal order to have high degree and nodes late in the causal order to have low degree implies that the variance explained by and into a node decreases along the causal order when $R^2$ is enforced to be constant. This results in early causal order nodes having many edges of very small effect. In our experiments, while the original simulations had a median per-edge variance explained of $0.025^2 \approx 0.063$, the effects of the root node have a median VE of just 0.004 (IQR: 0.002-0.008) after re-normalizing. This raises an important modeling question of whether early nodes in SF graphs should be expected to exhibit many extremely weak effects, and, if so, whether errors on such edges meaningfully affect downstream system behavior. We report the performance under this simulation in Table 4.

| Dim | Metric | IBCD | SDCD | GES | GIES | LiNGAM | igsp |
|---|---|---|---|---|---|---|---|
| 50D | F1 | 0.941±0.012 | 0.521±0.083 | 0.448±0.068 | 0.618±0.086 | 0.592±0.074 | 0.804±0.033 |
| | SHD | 28.2±7.8 | 165.9±36.9 | 334.3±78.1 | 211.4±66.7 | 221.9±63.7 | 95.5±21.6 |
| | SID | 373.2±96.2 | 1512.7±187.1 | 1209.2±161.0 | 952.1±163.3 | 1128.8±165.9 | 661.7±87.5 |
| | AUPRC | 0.982±0.006 | 0.388±0.062 | 0.257±0.051 | 0.424±0.092 | 0.491±0.081 | 0.667±0.045 |
| 250D | F1 | 0.389±0.047 | 0.374±0.023 | 0.528±0.056 | 0.539±0.058 | 0.449±0.058 | 0.810±0.036 |
| | SHD | 3585.5±856.9 | 1001.4±76.6 | 1618.5±334.5 | 1527.6±375.2 | 2049.8±334.5 | 508.0±120.3 |
| | SID | 10256.0±1558.0 | 32289.2±4858.5 | 12147.2±2812.7 | 9068.9±2092.2 | 12879.2±2562.9 | 9061.4±1141.7 |
| | AUPRC | 0.836±0.020 | 0.211±0.014 | 0.309±0.054 | 0.318±0.054 | 0.418±0.052 | 0.665±0.054 |

Table 4: Performance on scale-free graphs with reduced $R^2$-sortability via edge-weight reweighting.

For computing F1, SHD, and SID, we threshold edge estimates at the minimum nonzero value of $G'$, which is approximately 0.07 for 50 node graphs and decreases to approximately 0.01 for 250 node SF graphs. Under this setting, the performance of IBCD declines relative to the default simulations as our IV based approach would require extremely strong instruments to resolve tiny effects. However, IBCD achieves the highest AUPRC, indicating that pruning small estimates still gives good performance. GIES and igsp perform well here with strong SID, although IBCD remains very competitive with similar but slightly higher SID scores.

To mitigate the proliferation of extremely small effects, we additionally consider a second simulation in which the reweighted graph is thresholded by removing edges with absolute value below $v/2 = 0.125$, corresponding to half the minimum value of the original edge-weight distribution. This modification has two effects. It slows the decay of the degree distribution by disproportionately pruning edges from high-degree nodes, and it further reduces $R^2$-sortability, yielding $0.37 \pm 0.06$. This still results in high degree nodes having substantially smaller effects on average than low degree nodes. To evaluate SHD, F1 and SID, we return to our default threshold of 0.075 and present results below in Table 5.

| Dim | Metric | IBCD | SDCD | GES | GIES | LiNGAM | igsp |
|-----|--------|------|------|-----|------|--------|------|
| 50D | F1 | 0.965±0.014 | 0.588±0.086 | 0.474±0.102 | 0.649±0.067 | 0.618±0.164 | 0.856±0.045 |
| | SHD | 14.4±6.24 | 129.8±30.06 | 289.7±82.83 | 172.5±44.16 | 178.0±85.87 | 62.9±21.83 |
| | SID | 280.2±155.9 | 1486.1±174.4 | 1120.3±185.3 | 884.4±108.0 | 1121.8±422.1 | 608.0±113.5 |
| | AUPRC | 0.992±0.006 | 0.434±0.077 | 0.279±0.093 | 0.455±0.074 | 0.527±0.188 | 0.745±0.070 |
| 250D | F1 | 0.896±0.013 | 0.503±0.020 | 0.696±0.058 | 0.783±0.071 | 0.599±0.034 | 0.855±0.038 |
| | SHD | 178.1±24.65 | 600.5±38.72 | 659.5±193.05 | 440.7±190.87 | 783.5±81.72 | 271.9±79.59 |
| | SID | 7933.8±1140.4 | 30455.5±2605.3 | 12535.8±1955.6 | 8302.7±2654.5 | 25630.3±3162.1 | 5984.0±1917.5 |
| | AUPRC | 0.960±0.005 | 0.741±0.058 | 0.512±0.069 | 0.634±0.096 | 0.474±0.038 | 0.741±0.058 |

Table 5: Performance on thresholded scale-free graphs with reduced $R^2$-sortability.

Under this second simulation, IBCD again achieves strong performance, attaining the highest F1, SHD, and AUPRC across both graph sizes, while achieving SID comparable to igsp.

More broadly, these experiments highlight an interaction between eliminating sortability artifacts and preserving realistic effect-size distributions in scale-free linear SEMs. While non-sortable constructions are desirable from a benchmarking perspective, they may induce regimes in which errors on early order edges have limited downstream impact where an effect is not captured by existing metrics such as SID. Developing evaluation metrics that explicitly account for the functional consequences of small magnitude causal effects is an important direction for future work, but lies beyond the scope of the present study.

## G   Effect of soft acyclicity constraints on spectral radius $\rho_G$

Although IBCD does not require the inferred graph to be acyclic, we additionally evaluated a soft acyclicity constraint for completeness. Following LLCB (Weinstock et al., 2024), we placed a Gaussian prior on the spectral radius $\rho_G$ of the inferred graph, $\log \mathcal{N}(\rho_G; 0, \sigma_R^2) = -\frac{\rho_G^2}{2\sigma_R^2} + C$, which softly penalizes large spectral radius $\rho_G$ while allowing non-DAG structures. In practice, this constraint did not improve performance and made posterior optimization more difficult. We compare the resultant $\rho_G$ here to NOTEARS, which also uses a soft DAG constraint. As shown in Table 6, imposing a soft acyclicity constraint does not substantially change the spectral radius $\rho_G$ of graphs learned by IBCD. Compared to NOTEARS, IBCD produces graphs with slightly larger $\rho_G$ on ER graphs and slightly smaller $\rho_G$ on SF graphs. This concludes that our approach does not yield overly cyclic graphs relative to other soft-constrained methods.

| Dim | Graph | IBCD | IBCD(DAG) | NOTEARS |
|---|---|---|---|---|
| 50D | ER | $0.4487 \pm 0.0255$ | $0.4516 \pm 0.0222$ | $0.3073 \pm 0.0212$ |
| | SF | $0.1697 \pm 0.0242$ | $0.1752 \pm 0.0323$ | $0.3470 \pm 0.1083$ |
| 150D | ER | $0.4314 \pm 0.0219$ | $0.4721 \pm 0.1245$ | $0.2948 \pm 0.0205$ |
| | SF | $0.2377 \pm 0.0232$ | $0.2392 \pm 0.0192$ | $0.4173 \pm 0.1068$ |
| 250D | ER | $0.4269 \pm 0.0143$ | $0.4312 \pm 0.0181$ | $0.2790 \pm 0.0189$ |
| | SF | $0.2605 \pm 0.0149$ | $0.2805 \pm 0.0345$ | $0.5166 \pm 0.0812$ |
| 500D | ER | $0.4095 \pm 0.0161$ | $0.4111 \pm 0.0188$ | NA |
| | SF | $0.3498 \pm 0.0300$ | $0.3443 \pm 0.0278$ | NA |

Table 6: Comparison of the mean and standard deviation of the spectral radius $\rho_G$ with increasing numbers of dimensions on ER and SF graphs with and without a soft acyclicity constraint. NA values correspond to failed runs, typically due to a 12 hour timeout or termination for exceeding resource limits as noted in Table 1.

# H  Additional figures and tables

| Graph | Metric | IBCD | BayesDAG | GIES | GES |
|---|---|---|---|---|---|
| ER | E-F1 | 0.814±0.015 | 0.145±0.008 | 0.538±0.123 | 0.328±0.036 |
| | E-SHD | 110.2±12.43 | 737.0±7.88 | 394.5±118.15 | 636.1±88.70 |
| SF | E-F1 | 0.932±0.008 | 0.144±0.013 | 0.598±0.037 | 0.447±0.039 |
| | E-SHD | 33.1±4.88 | 730.8±14.61 | 236.9±41.16 | 340.4±42.82 |

Table 7: E-F1 and SHD on ER and SF graphs for GES, GIES, IBCD and BayesDAG ($D = 50, N = 100$).

| Dim | Graph | IBCD | inspre | GIES | GES | LiNGAM | SDCD | NOTEARS | BayesDAG | LLCB | igsp |
|---|---|---|---|---|---|---|---|---|---|---|---|
| 50D | ER | $0.971 \pm 0.008$ | $0.921 \pm 0.018$ | $0.536 \pm 0.129$ | $0.329 \pm 0.054$ | $0.344 \pm 0.058$ | $0.422 \pm 0.102$ | $0.559 \pm 0.046$ | $0.400 \pm 0.040$ | $0.408 \pm 0.037$ | $0.490 \pm 0.064$ |
| | SF | $0.965 \pm 0.011$ | $0.886 \pm 0.020$ | $0.617 \pm 0.074$ | $0.451 \pm 0.042$ | $0.801 \pm 0.107$ | $0.642 \pm 0.065$ | $0.474 \pm 0.050$ | $0.483 \pm 0.086$ | $0.426 \pm 0.077$ | $0.825 \pm 0.033$ |
| 150D | ER | $0.978 \pm 0.004$ | $0.893 \pm 0.008$ | $0.638 \pm 0.119$ | $0.385 \pm 0.066$ | $0.285 \pm 0.022$ | $0.300 \pm 0.040$ | $0.801 \pm 0.027$ | NA | NA | $0.351 \pm 0.032$ |
| | SF | $0.959 \pm 0.009$ | $0.808 \pm 0.011$ | $0.520 \pm 0.059$ | $0.447 \pm 0.027$ | $0.841 \pm 0.072$ | $0.582 \pm 0.042$ | $0.537 \pm 0.029$ | NA | NA | $0.816 \pm 0.028$ |
| 250D | ER | $0.979 \pm 0.003$ | $0.879 \pm 0.006$ | $0.710 \pm 0.140$ | $0.443 \pm 0.063$ | $0.276 \pm 0.026$ | $0.311 \pm 0.029$ | $0.857 \pm 0.010$ | NA | NA | $0.261 \pm 0.023$ |
| | SF | $0.963 \pm 0.004$ | $0.785 \pm 0.011$ | $0.492 \pm 0.033$ | $0.441 \pm 0.028$ | $0.836 \pm 0.040$ | $0.597 \pm 0.036$ | $0.551 \pm 0.028$ | NA | NA | $0.781 \pm 0.035$ |
| 500D | ER | $0.982 \pm 0.003$ | $0.851 \pm 0.010$ | $0.789 \pm 0.110$ | $0.590 \pm 0.076$ | $0.280 \pm 0.031$ | $0.307 \pm 0.016$ | NA | NA | NA | $0.198 \pm 0.016$ |
| | SF | $0.943 \pm 0.015$ | $0.738 \pm 0.010$ | $0.447 \pm 0.025$ | $0.437 \pm 0.027$ | $0.856 \pm 0.029$ | $0.659 \pm 0.019$ | NA | NA | NA | $0.683 \pm 0.026$ |

Table 8: Comparison of mean and standard deviation of F1 with increasing numbers of dimensions on ER and SF graphs in Figure 2. NA values correspond to failed runs, typically due to a 12 hour timeout or termination for exceeding resource limits as noted in Table 1.

| Dim | Graph | IBCD | inspre | GIES | GES | LiNGAM | SDCD | NOTEARS | BayesDAG | LLCB | igsp |
|---|---|---|---|---|---|---|---|---|---|---|---|
| 50D | ER | $14.8 \pm 4.26$ | $41.3 \pm 11.09$ | $374.6 \pm 168.58$ | $632.1 \pm 106.99$ | $510.8 \pm 82.95$ | $219.9 \pm 44.66$ | $184.3 \pm 24.53$ | $391.7 \pm 19.52$ | $640.8 \pm 114.74$ | $372.8 \pm 85.94$ |
| | SF | $16.6 \pm 5.76$ | $56.0 \pm 12.87$ | $218.0 \pm 62.85$ | $329.0 \pm 49.67$ | $101.9 \pm 55.63$ | $137.2 \pm 30.98$ | $196.6 \pm 32.09$ | $287.4 \pm 58.90$ | $316.4 \pm 41.47$ | $85.9 \pm 20.07$ |
| 150D | ER | $32.0 \pm 6.99$ | $163.4 \pm 15.15$ | $829.7 \pm 375.50$ | $1958.8 \pm 472.12$ | $1889.4 \pm 145.69$ | $695.0 \pm 40.04$ | $266.3 \pm 34.63$ | NA | NA | $1993.3 \pm 243.69$ |
| | SF | $61.2 \pm 13.65$ | $279.7 \pm 24.39$ | $1024.9 \pm 214.49$ | $1247.2 \pm 143.42$ | $261.6 \pm 143.87$ | $462.8 \pm 57.69$ | $543.9 \pm 53.11$ | NA | NA | $299.9 \pm 55.84$ |
| 250D | ER | $53.5 \pm 8.58$ | $308.9 \pm 23.67$ | $1094.3 \pm 762.81$ | $2819.5 \pm 739.89$ | $3254.1 \pm 317.28$ | $1178.5 \pm 58.18$ | $329.0 \pm 20.62$ | NA | NA | $5015.4 \pm 587.11$ |
| | SF | $91.7 \pm 10.80$ | $512.0 \pm 43.91$ | $1927.5 \pm 261.34$ | $2247.6 \pm 328.42$ | $435.5 \pm 118.99$ | $731.6 \pm 85.21$ | $862.1 \pm 90.74$ | NA | NA | $610.4 \pm 119.74$ |
| 500D | ER | $91.5 \pm 12.79$ | $707.9 \pm 28.25$ | $1417.9 \pm 1009.78$ | $3348.2 \pm 980.04$ | $6306.1 \pm 704.84$ | $2638.6 \pm 83.47$ | NA | NA | NA | $14190.4 \pm 1362.75$ |
| | SF | $284.8 \pm 74.40$ | $1316.5 \pm 37.60$ | $4831.8 \pm 434.00$ | $4972.1 \pm 536.01$ | $782.1 \pm 190.47$ | $1319.3 \pm 88.99$ | NA | NA | NA | $2024.5 \pm 252.74$ |

Table 9: Comparison of mean and standard deviation of SHD with increasing numbers of dimensions on ER and SF graphs in Figure 2. NA values correspond to failed runs, typically due to a 12 hour timeout or termination for exceeding resource limits as noted in Table 1.

| Dim | Graph | IBCD | inspre | GIES | GES | LiNGAM | SDCD | NOTEARS | BayesDAG | igsp |
|-----|-------|------|--------|------|-----|--------|------|---------|----------|------|
| 50D | ER | 489.0±161.56 | 892.4±221.74 | 1277.3±215.88 | 1771.3±155.41 | 1945.9±108.32 | 2212.3±89.34 | 2168.5±101.14 | 2126.1±69.59 | 1686.9±96.86 |
|     | SF | 297.0±84.20 | 937.3±159.24 | 916.4±163.54 | 1121.0±78.26 | 697.4±285.21 | 1481.0±184.60 | 1700.6±134.17 | 1597.1±190.64 | 684.6±110.00 |
| 150D | ER | 3450.8±633.14 | 9545.2±612.09 | 8171.8±2569.26 | 14270.3±1418.95 | 19822.3±337.74 | 20382.5±439.61 | 15839.4±953.84 | NA | 15252.2±595.63 |
|      | SF | 3257.8±827.84 | 11587.0±716.01 | 6675.4±403.03 | 6663.6±1102.50 | 5432.6±1171.86 | 13312.2±742.52 | 13297.7±1077.35 | NA | 4263.5±560.35 |
| 250D | ER | 8421.3±1306.56 | 28569.1±1696.13 | 16657.0±8727.20 | 34510.2±3156.43 | 56919.9±627.81 | 57262.3±1232.99 | 36444.5±1741.38 | NA | 42728.5±1418.92 |
|      | SF | 7658.1±955.31 | 29741.3±2398.19 | 12526.9±1732.78 | 13271.6±2345.28 | 13565.9±3193.94 | 29798.5±3619.30 | 28637.5±4760.00 | NA | 9296.4±1032.59 |
| 500D | ER | 11785.7±1917.27 | 32304.0±4034.13 | 12418.3±7627.63 | 37246.3±5659.40 | 67790.4±5739.75 | 76201.0±4450.63 | NA | NA | 59775.7±3700.01 |
|      | SF | 2095.8±828.30 | NA | 11064.2±1215.04 | 13661.3±2549.48 | 5967.6±1626.90 | 14315.5±2020.22 | NA | NA | 10503.4±1327.20 |

Table 10: Comparison of mean and standard deviation of SID with increasing numbers of dimensions on ER and SF graph in Figure 2. NA values correspond to failed runs, typically due to a 12 hour timeout or termination for exceeding resource limits as noted in Table 1.

| Dim | Graph | IBCD | inspre | GIES | GES | LiNGAM | SDCD | NOTEARS | BayesDAG | LLCB | igsp |
|-----|-------|------|--------|------|-----|--------|------|---------|----------|------|------|
| 50D | ER | 0.995±0.007 | 0.983±0.008 | 0.356±0.122 | 0.180±0.032 | 0.253±0.056 | 0.275±0.074 | 0.579±0.045 | 0.219±0.029 | 0.438±0.032 | 0.299±0.052 |
|     | SF | 0.983±0.007 | 0.950±0.015 | 0.422±0.079 | 0.256±0.032 | 0.699±0.143 | 0.494±0.065 | 0.438±0.048 | 0.288±0.072 | 0.312±0.021 | 0.697±0.048 |
| 150D | ER | 0.993±0.009 | 0.959±0.008 | 0.463±0.133 | 0.213±0.049 | 0.185±0.026 | 0.146±0.029 | 0.849±0.023 | NA | NA | 0.178±0.022 |
|      | SF | 0.978±0.010 | 0.858±0.018 | 0.307±0.055 | 0.236±0.023 | 0.778±0.098 | 0.409±0.037 | 0.476±0.032 | NA | NA | 0.674±0.041 |
| 250D | ER | 0.996±0.001 | 0.940±0.009 | 0.556±0.162 | 0.263±0.050 | 0.177±0.023 | 0.139±0.020 | 0.909±0.010 | NA | NA | 0.119±0.013 |
|      | SF | 0.984±0.003 | 0.830±0.013 | 0.282±0.029 | 0.234±0.023 | 0.785±0.048 | 0.419±0.035 | 0.489±0.033 | NA | NA | 0.622±0.049 |
| 500D | ER | 0.996±0.001 | 0.901±0.018 | 0.657±0.134 | 0.408±0.077 | 0.176±0.029 | 0.112±0.010 | NA | NA | NA | 0.084±0.008 |
|      | SF | 0.971±0.017 | 0.792±0.011 | 0.247±0.021 | 0.238±0.023 | 0.821±0.037 | 0.479±0.020 | NA | NA | NA | 0.491±0.032 |

Table 11: Comparison of mean and standard deviation of AUPRC with increasing numbers of dimensions on ER and SF graph in Figure 2. NA values correspond to failed runs, typically due to a 12 hour timeout or termination for exceeding resource limits as noted in Table 1.

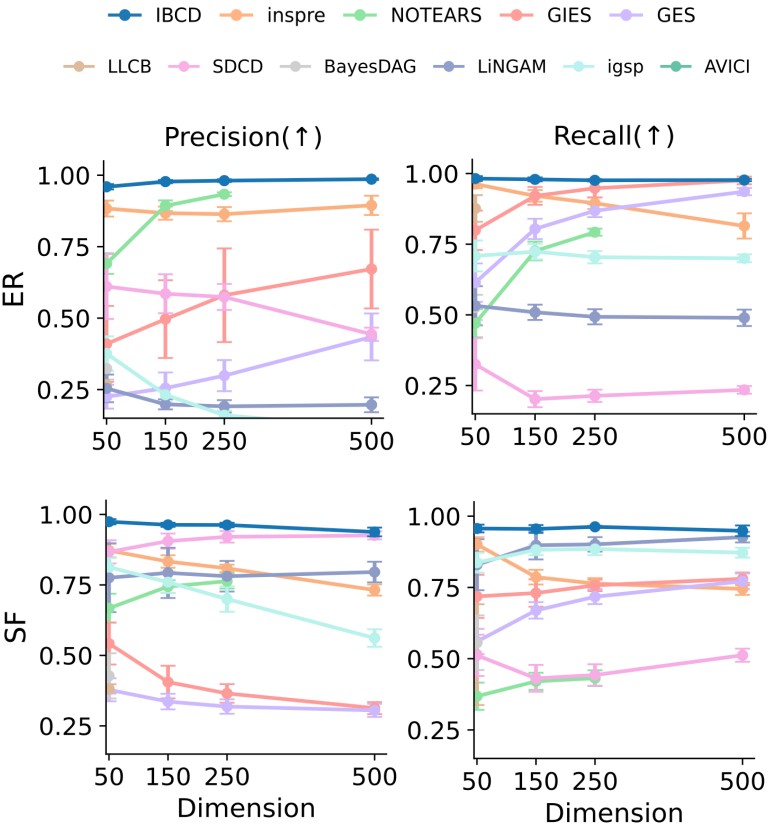

Figure 6: Comparison of precision and recall with increasing numbers of dimensions on ER and SF graphs, using 100 intervention samples per variable.

| Sample | Graph | IBCD | inspre | GIES | GES | LiNGAM | SDCD | NOTEARS | BayesDAG | LLCB | igsp | AVICI |
|---|---|---|---|---|---|---|---|---|---|---|---|---|
| 5 | ER | 0.431 ± 0.030 | 0.311 ± 0.030 | 0.642 ± 0.077 | 0.434 ± 0.071 | 0.299 ± 0.054 | 0.737 ± 0.036 | 0.499 ± 0.055 | 0.007 ± 0.009 | 0.847 ± 0.023 | 0.259 ± 0.032 | 0.253 ± 0.042 |
|  | SF | 0.518 ± 0.054 | 0.279 ± 0.037 | 0.536 ± 0.036 | 0.452 ± 0.049 | 0.522 ± 0.078 | 0.622 ± 0.065 | 0.443 ± 0.049 | 0.002 ± 0.004 | 0.633 ± 0.043 | 0.416 ± 0.055 | 0.234 ± 0.114 |
| 15 | ER | 0.708 ± 0.035 | 0.517 ± 0.029 | 0.622 ± 0.075 | 0.394 ± 0.079 | 0.364 ± 0.054 | 0.684 ± 0.059 | 0.535 ± 0.058 | 0.220 ± 0.058 | 0.872 ± 0.026 | 0.477 ± 0.047 | 0.256 ± 0.050 |
|  | SF | 0.784 ± 0.019 | 0.496 ± 0.035 | 0.592 ± 0.072 | 0.498 ± 0.044 | 0.760 ± 0.092 | 0.737 ± 0.051 | 0.471 ± 0.045 | 0.227 ± 0.070 | 0.586 ± 0.128 | 0.691 ± 0.041 | 0.276 ± 0.122 |
| 25 | ER | 0.813 ± 0.026 | 0.630 ± 0.024 | 0.609 ± 0.137 | 0.387 ± 0.053 | 0.332 ± 0.061 | 0.628 ± 0.059 | 0.542 ± 0.057 | 0.355 ± 0.048 | 0.882 ± 0.025 | 0.525 ± 0.037 | 0.244 ± 0.040 |
|  | SF | 0.868 ± 0.022 | 0.589 ± 0.032 | 0.640 ± 0.049 | 0.476 ± 0.078 | 0.789 ± 0.077 | 0.740 ± 0.046 | 0.462 ± 0.050 | 0.390 ± 0.070 | 0.564 ± 0.161 | 0.741 ± 0.043 | 0.276 ± 0.120 |
| 50 | ER | 0.913 ± 0.014 | 0.779 ± 0.018 | 0.553 ± 0.116 | 0.334 ± 0.042 | 0.356 ± 0.041 | 0.513 ± 0.085 | 0.549 ± 0.052 | 0.426 ± 0.037 | 0.891 ± 0.025 | 0.520 ± 0.051 | 0.226 ± 0.033 |
|  | SF | 0.934 ± 0.011 | 0.749 ± 0.013 | 0.647 ± 0.067 | 0.469 ± 0.060 | 0.786 ± 0.083 | 0.703 ± 0.048 | 0.458 ± 0.054 | 0.504 ± 0.078 | 0.573 ± 0.146 | 0.800 ± 0.043 | 0.295 ± 0.127 |
| 75 | ER | 0.950 ± 0.018 | 0.867 ± 0.023 | 0.534 ± 0.135 | 0.344 ± 0.053 | 0.338 ± 0.057 | 0.461 ± 0.119 | 0.563 ± 0.063 | 0.433 ± 0.049 | 0.878 ± 0.043 | 0.515 ± 0.063 | NA |
|  | SF | 0.953 ± 0.013 | 0.827 ± 0.027 | 0.649 ± 0.036 | 0.481 ± 0.050 | 0.828 ± 0.072 | 0.694 ± 0.051 | 0.476 ± 0.062 | 0.538 ± 0.051 | 0.532 ± 0.166 | 0.798 ± 0.049 | NA |
| 100 | ER | 0.971 ± 0.008 | 0.921 ± 0.018 | 0.536 ± 0.129 | 0.329 ± 0.054 | 0.344 ± 0.058 | 0.422 ± 0.102 | 0.559 ± 0.046 | 0.400 ± 0.040 | 0.876 ± 0.047 | 0.490 ± 0.064 | NA |
|  | SF | 0.965 ± 0.011 | 0.886 ± 0.020 | 0.617 ± 0.074 | 0.451 ± 0.042 | 0.801 ± 0.107 | 0.642 ± 0.065 | 0.474 ± 0.050 | 0.483 ± 0.086 | 0.514 ± 0.177 | 0.825 ± 0.033 | NA |

Table 12: Comparison of mean and standard deviation of F1 with increasing intervention sample sizes on ER and SF graphs with $D = 50$ in Figure 3. NA values correspond to failed runs, typically due to a 12 hour timeout or termination for exceeding resource limits as noted in Table 1.

| Sample | Graph | IBCD | inspre | GIES | GES | LiNGAM | SDCD | NOTEARS | BayesDAG | LLCB | igsp | AVICI |
|---|---|---|---|---|---|---|---|---|---|---|---|---|
| 5 | ER | 225.3 ± 22.72 | 480.7 ± 71.30 | 203.5 ± 55.72 | 348.4 ± 62.84 | 304.6 ± 36.67 | 128.5 ± 21.66 | 208.7 ± 30.08 | 248.7 ± 16.57 | 1024.6 ± 48.99 | 323.1 ± 23.01 | 422.3 ± 68.64 |
|  | SF | 186.9 ± 23.59 | 423.0 ± 85.45 | 225.3 ± 34.65 | 268.3 ± 36.97 | 183.5 ± 36.10 | 167.9 ± 34.80 | 206.8 ± 31.47 | 241.2 ± 27.53 | 891.3 ± 23.32 | 206.2 ± 33.90 | 488.3 ± 178.68 |
| 15 | ER | 135.8 ± 19.32 | 280.3 ± 28.10 | 243.2 ± 77.52 | 452.8 ± 103.42 | 452.0 ± 73.57 | 141.5 ± 29.49 | 194.6 ± 29.35 | 253.9 ± 21.92 | 759.6 ± 89.35 | 299.4 ± 37.60 | 382.6 ± 49.88 |
|  | SF | 96.3 ± 14.67 | 244.7 ± 43.44 | 214.3 ± 58.74 | 269.1 ± 43.67 | 116.1 ± 48.42 | 110.9 ± 25.36 | 198.6 ± 29.78 | 230.8 ± 29.37 | 483.9 ± 81.67 | 133.8 ± 28.40 | 352.6 ± 68.48 |
| 25 | ER | 93.3 ± 16.42 | 225.2 ± 19.60 | 271.2 ± 136.74 | 484.6 ± 81.31 | 514.0 ± 85.27 | 155.3 ± 27.92 | 191.3 ± 30.22 | 261.1 ± 23.30 | 702.8 ± 92.57 | 291.5 ± 37.38 | 372.5 ± 67.68 |
|  | SF | 59.3 ± 7.59 | 212.7 ± 29.58 | 190.4 ± 42.08 | 289.1 ± 67.28 | 103.3 ± 37.02 | 108.8 ± 25.94 | 202.0 ± 33.13 | 223.6 ± 37.25 | 389.2 ± 76.47 | 120.3 ± 31.61 | 345.6 ± 72.38 |
| 50 | ER | 44.2 ± 8.28 | 126.0 ± 17.38 | 338.0 ± 132.32 | 595.6 ± 85.51 | 500.3 ± 71.75 | 192.6 ± 35.68 | 188.6 ± 27.82 | 303.5 ± 30.78 | 672.2 ± 87.79 | 325.0 ± 65.56 | 342.7 ± 49.07 |
|  | SF | 30.5 ± 4.25 | 126.8 ± 16.64 | 193.5 ± 58.40 | 302.6 ± 56.32 | 109.5 ± 44.46 | 120.9 ± 25.94 | 200.2 ± 31.07 | 232.9 ± 48.99 | 350.0 ± 42.71 | 96.7 ± 26.43 | 324.3 ± 58.55 |
| 75 | ER | 25.6 ± 8.68 | 72.5 ± 14.08 | 370.4 ± 154.47 | 594.0 ± 103.28 | 518.6 ± 73.58 | 209.4 ± 52.91 | 184.4 ± 32.34 | 342.6 ± 35.45 | 639.1 ± 113.90 | 341.2 ± 68.31 | NA |
|  | SF | 22.2 ± 6.07 | 86.1 ± 12.05 | 191.9 ± 36.30 | 301.3 ± 51.68 | 85.4 ± 35.29 | 122.2 ± 27.58 | 196.8 ± 36.70 | 238.4 ± 37.65 | 330.6 ± 39.50 | 100.4 ± 35.19 | NA |
| 100 | ER | 14.8 ± 4.26 | 41.3 ± 11.09 | 374.6 ± 168.58 | 632.1 ± 106.99 | 510.8 ± 82.95 | 219.9 ± 44.66 | 184.3 ± 24.53 | 391.7 ± 19.52 | 640.8 ± 114.74 | 372.8 ± 85.94 | NA |
|  | SF | 16.6 ± 5.76 | 56.0 ± 12.87 | 218.0 ± 62.85 | 329.0 ± 49.67 | 101.9 ± 55.63 | 137.2 ± 30.98 | 196.6 ± 32.09 | 287.4 ± 58.90 | 316.4 ± 41.47 | 85.9 ± 20.07 | NA |

Table 13: Comparison of mean and standard deviation of SHD with increasing intervention sample sizes on ER and SF graphs with $D = 50$ in Figure 3. NA values correspond to failed runs, typically due to a 12 hour timeout or termination for exceeding resource limits as noted in Table 1.

| Sample | Graph | IBCD | inspre | GIES | GES | LiNGAM | SDCD | NOTEARS | BayesDAG | igsp | AVICI |
|---|---|---|---|---|---|---|---|---|---|---|---|
| 5 | ER | 2285.8±63.69 | 2207.0±36.10 | 1734.0±231.27 | 2019.6±99.73 | 2229.2±66.28 | 1612.8±194.79 | 2239.7±76.36 | 2138.9±114.74 | 2297.5±54.37 | 1853.3±118.60 |
|  | SF | 1956.3±116.28 | 2189.0±43.28 | 1365.7±151.58 | 1480.0±144.07 | 2124.0±191.38 | 1415.7±185.12 | 1776.3±119.55 | 2392.6±2.07 | 2228.0±81.80 | 2016.7±229.71 |
| 15 | ER | 2048.6±112.26 | 2153.4±50.11 | 1527.3±184.32 | 1942.2±131.11 | 2038.5±70.46 | 1838.4±190.91 | 2210.4±108.98 | 2258.0±72.59 | 2030.7±72.39 | 1869.8±150.58 |
|  | SF | 1376.9±148.91 | 1970.0±77.78 | 1054.5±145.81 | 1212.6±131.24 | 1233.4±333.76 | 1194.5±123.89 | 1717.7±120.65 | 2294.4±46.92 | 1703.4±199.90 | 2022.2±240.38 |
| 25 | ER | 1759.8±149.39 | 2027.9±93.06 | 1350.8±283.18 | 1890.1±127.14 | 1966.7±94.95 | 1976.1±108.15 | 2226.7±78.88 | 2266.0±77.72 | 1940.2±96.48 | 1900.3±142.67 |
|  | SF | 1026.7±126.44 | 1703.4±133.76 | 984.7±69.66 | 1236.8±142.07 | 973.7±188.59 | 1204.4±207.71 | 1712.4±117.88 | 2094.1±103.53 | 1344.3±214.90 | 2039.8±282.16 |
| 50 | ER | 1087.1±182.61 | 1577.6±93.21 | 1351.1±255.23 | 1857.7±112.13 | 1927.3±89.98 | 2137.6±149.23 | 2182.8±113.62 | 2202.1±62.27 | 1776.5±130.78 | 1952.0±142.26 |
|  | SF | 641.8±84.20 | 1470.4±143.45 | 911.3±112.73 | 1146.9±116.01 | 761.9±199.64 | 1291.5±195.29 | 1704.4±97.38 | 1764.1±186.72 | 867.0±180.10 | 1984.9±315.30 |
| 75 | ER | 664.9±169.68 | 1161.4±143.85 | 1291.4±348.25 | 1794.8±126.11 | 1929.1±104.93 | 2179.9±115.93 | 2166.9±112.80 | 2143.0±70.26 | 1694.9±126.88 | NA |
|  | SF | 389.8±111.79 | 1175.0±98.41 | 904.4±79.72 | 1110.6±84.34 | 686.6±251.36 | 1338.1±184.12 | 1690.7±136.45 | 1582.1±188.68 | 791.0±160.24 | NA |
| 100 | ER | 489.0±161.56 | 892.4±221.74 | 1277.3±215.88 | 1771.3±155.41 | 1945.9±108.32 | 2212.3±89.34 | 2168.5±101.14 | 2126.1±69.59 | 1686.9±96.86 | NA |
|  | SF | 297.0±84.20 | 937.3±159.24 | 916.4±163.54 | 1121.0±78.26 | 697.4±285.21 | 1481.0±184.60 | 1700.6±134.17 | 1597.1±190.64 | 684.6±110.00 | NA |

Table 14: Comparison of mean and standard deviation of SID with increasing intervention sample sizes on ER and SF graphs with $D = 50$ in Figure 3. NA values correspond to failed runs, typically due to a 12 hour timeout or termination for exceeding resource limits as noted in Table 1.

| Sample | Graph | IBCD | inspre | GIES | GES | LiNGAM | SDCD | NOTEARS | BayesDAG | LLCB | igsp | AVICI |
|---|---|---|---|---|---|---|---|---|---|---|---|---|
| 5 | ER | 0.434±0.038 | 0.279±0.018 | 0.451±0.084 | 0.247±0.051 | 0.169±0.032 | 0.573±0.047 | 0.477±0.038 | 0.104±0.006 | 0.366±0.024 | 0.148±0.012 | 0.232±0.039 |
|  | SF | 0.600±0.071 | 0.260±0.010 | 0.334±0.030 | 0.260±0.036 | 0.348±0.087 | 0.435±0.063 | 0.400±0.028 | 0.099±0.011 | 0.154±0.011 | 0.271±0.042 | 0.198±0.097 |
| 15 | ER | 0.765±0.038 | 0.568±0.047 | 0.432±0.074 | 0.221±0.052 | 0.242±0.049 | 0.517±0.067 | 0.537±0.056 | 0.156±0.019 | 0.426±0.030 | 0.279±0.038 | 0.241±0.043 |
|  | SF | 0.870±0.012 | 0.470±0.025 | 0.393±0.071 | 0.296±0.036 | 0.641±0.133 | 0.591±0.066 | 0.432±0.038 | 0.170±0.038 | 0.244±0.031 | 0.523±0.045 | 0.250±0.119 |
| 25 | ER | 0.884±0.018 | 0.739±0.028 | 0.430±0.140 | 0.215±0.035 | 0.227±0.052 | 0.461±0.055 | 0.555±0.063 | 0.207±0.028 | 0.433±0.027 | 0.323±0.033 | 0.243±0.039 |
|  | SF | 0.937±0.013 | 0.635±0.051 | 0.445±0.054 | 0.281±0.063 | 0.675±0.105 | 0.599±0.055 | 0.424±0.046 | 0.242±0.041 | 0.280±0.039 | 0.581±0.052 | 0.245±0.110 |
| 50 | ER | 0.972±0.006 | 0.908±0.018 | 0.370±0.116 | 0.181±0.024 | 0.249±0.034 | 0.348±0.075 | 0.570±0.059 | 0.239±0.024 | 0.440±0.027 | 0.322±0.043 | 0.226±0.026 |
|  | SF | 0.973±0.005 | 0.830±0.018 | 0.455±0.072 | 0.272±0.047 | 0.683±0.120 | 0.555±0.052 | 0.434±0.055 | 0.310±0.063 | 0.302±0.022 | 0.662±0.060 | 0.264±0.119 |
| 75 | ER | 0.990±0.006 | 0.965±0.011 | 0.355±0.133 | 0.189±0.033 | 0.241±0.046 | 0.310±0.093 | 0.580±0.063 | 0.244±0.035 | 0.435±0.033 | 0.320±0.055 | NA |
|  | SF | 0.980±0.010 | 0.911±0.022 | 0.455±0.040 | 0.282±0.040 | 0.731±0.099 | 0.548±0.055 | 0.438±0.056 | 0.335±0.047 | 0.314±0.025 | 0.659±0.067 | NA |
| 100 | ER | 0.995±0.007 | 0.983±0.008 | 0.356±0.122 | 0.180±0.032 | 0.253±0.056 | 0.275±0.074 | 0.579±0.045 | 0.219±0.029 | 0.438±0.032 | 0.299±0.052 | NA |
|  | SF | 0.983±0.007 | 0.950±0.015 | 0.422±0.079 | 0.256±0.032 | 0.699±0.143 | 0.494±0.065 | 0.438±0.048 | 0.288±0.072 | 0.312±0.021 | 0.697±0.048 | NA |

Table 15: Comparison of mean and standard deviation of AUPRC with increasing intervention sample sizes on ER and SF graphs with $D = 50$ in Figure 3. NA values correspond to failed runs, typically due to a 12 hour timeout or termination for exceeding resource limits as noted in Table 1.

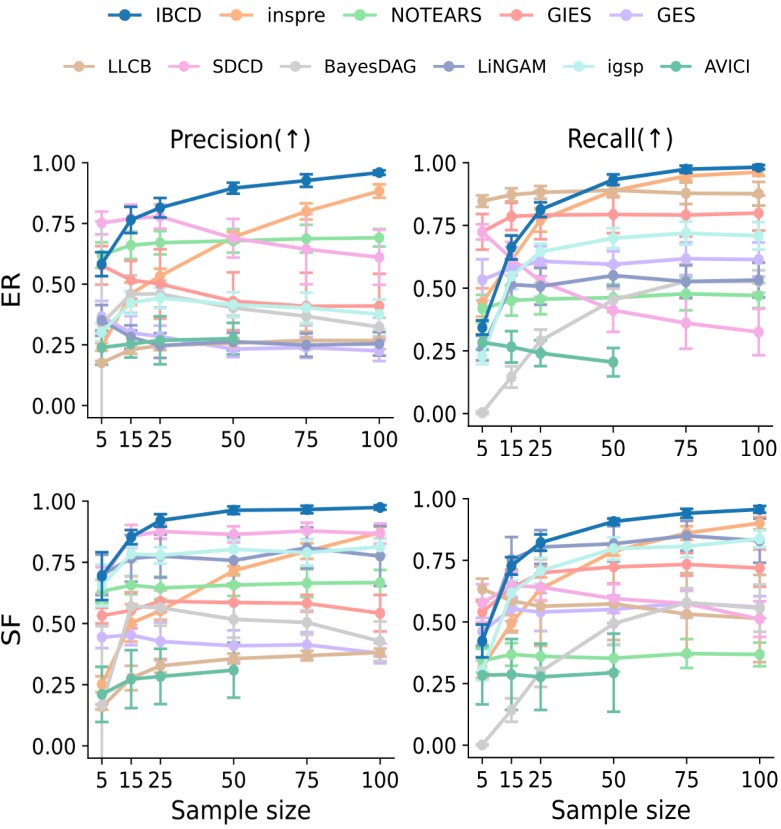

Figure 7: Comparison of precision and recall with increasing numbers of intervention sample sizes on ER and SF graphs with $D = 50$.

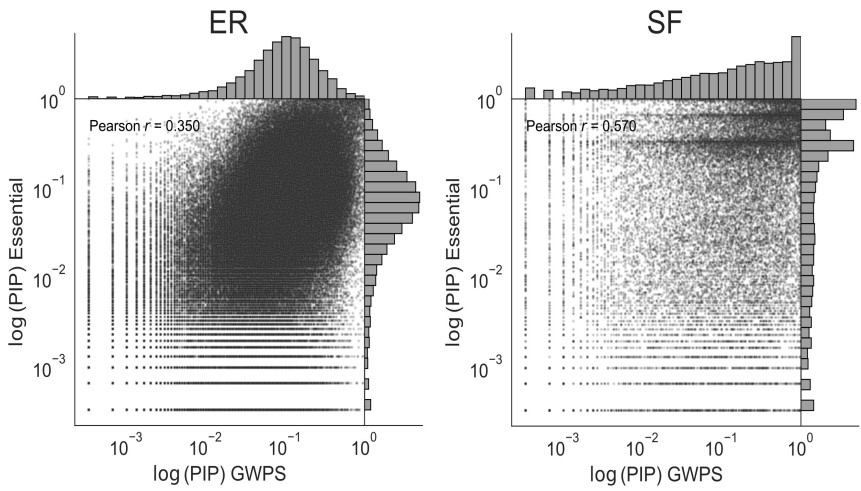

Figure 8: Agreement of log PIP values between GWPS and Essential K562 screens under ER and SF priors. Each point is one edge and darker regions indicate higher density. The SF prior shows stronger cross screen consistency.

