# OpenReview forum: "Large Scale Empirical Bayesian Causal Discovery Using Total Effect Estimates From Intervention Data"
_TMLR — Under review for TMLR_

### Review · Reviewer_gwET · 2026-06-17

**Summary Of Contributions:**

This paper proposes a method IBCD for estimating posterior distributions of Causal Bayesian Networks from observational and international datasets. Assumptions for identifiability include a linear causal model, and that interventions are modeled as instrumental variables on each other variable; this allows for a closed form solution of the average causal effect of an intervention using two-staged least squares. Instead of estimating the causal parents of each variable, this paper reframes causal discovery as direct effect estimation via this computed ACE. Additionally, IBCD yields full MCMC posterior samples of the causal graph. The authors show that the proposed algorithm outperforms baselines on synthetic data generated from Erdos Renyi and Scale free networks, as well as evaluation on real-world genomics data.

Pros:
- Strong empirical results with synthetic data against a wide range of commonly used baselines, shows that IBCD outperforms in both accuracy of recovered graph and compute time (for example faster than closest competitor igsp)
- Empirical results on calibrated posterior using PIP strengthens the results.

Cons:
- Real data results are based on correlation of PIP values for each edge across folds, this measures the edge stability of the algorithm but does not really measure the accuracy of the learned graphs.

**Additional Comments:**

Minor issues:
- End of the first paragraph on p1, Chickering citation does not have a year
- Confused by notation first paragraph of p3. Is i an intervention in {I_m}? Then what does x_i correspond to? Based on preceding notation this would map to a sample between 1 and N.
- Figure 1, explain $\lambda$
- Section 4.3 refers to content only in the appendix which was confusing. I would suggest either moving this text to the appendix with a briefer mention of the result or moving Table 7 to the main text.
- Can the authors clarify which other baselines sample the posterior? Table 7 only includes BayesDAG, GES and GIES, the latter two I believe do not believe produce posterior samples?
- Does the IBCD assume a Gaussian distribution?

**Audience:**

Yes

**Audience Explanation:**

This work is relevant to the causal discovery community. The paper shows an effective solution to a challenging problem of causal discovery at scale, and further of sampling posterior distributions of causal graphs.

**Broader Impact Concerns:**

No ethical concerns.

**Claims And Evidence:**

Yes

**Claims Explanation:**

I think this is a strong paper with a sound theoretical framework and very convincing empirical results on synthetic data. The theory is limited by the linear assumption, and requirement for interventions at IVs; however, the empirical results show IBCD outperforms baselines with the same/similar assumptions. Further the generation of the posterior distribution of graphs is always favored in real world settings. The empirical results are limited in the real data case which I think the authors should address (see below).

**Requested Changes:**

- I think the authors should include some kind of accuracy analysis for the Perturb-seq data. I understand a ground truth graph is difficult to obtain for here, however, the authors could follow the framework from CausalBench (https://www.nature.com/articles/s42003-025-07764-y) which evaluates learned graphs to knowledge bases. Additionally, I think some partial edge-wise knowledge is possible with respect to the intervened genes, for example any genes that are affected by a gene knockdown cannot be upstream of the interred gene. Some type of stability analysis across folds for this would be good.

---

> ### Author Response · Authors · 2026-07-08
>
> Thank you for the helpful suggestions. We agree that comparison to baselines on real perturb-seq data is important. Because a ground truth causal graph is unavailable and IBCD requires valid intervention coverage for each modeled variable, we added a complementary analysis on the 521 gene essential screen, where this assumption is satisfied. Specifically, we performed 5 fold stratified splitting, fit each method on the 80% training split of each fold, and assessed fold to fold consistency across the resulting five inferred graphs using SHD and F1 over all 10 graph pairs after thresholding edge magnitudes. IBCD was competitive with the strongest baselines with SHD 4,680.2±186.7, F1 0.630±0.01, compared with GIES 16,713.2±587.2, 0.698±0.01, igsp 4,352.6±70.9, 0.365±0.01, and SDCD 29.6±4.9, 0.678±0.05. We also note that SDCD’s extremely small SHD appears to reflect very sparse solutions across folds, which results in consistent but uninformative graphs. SID is not reported here because it requires a ground truth DAG, which is unavailable in the essential Perturb-seq setting.
>
> We also evaluated cross-validation reproducibility of posterior edge confidence on the essential screen, complementing the analysis in section 4.6. Bayesian baselines did not scale to the 521 gene setting (table 1) and most other baseline methods return only a single graph estimate, so obtaining edge inclusion probabilities requires computationally expensive bootstrap procedures. We therefore focused on the fast methods igsp and GIES. Using 100 bootstrap resamples on the 521-gene essential screen, igsp achieved a cross-fold PIP correlation of 0.776, compared with 0.817 for IBCD. Unfortunately in the case of GIES on the essential screen, a single fold took nearly 11 hours, so 100 bootstrap runs would require on the order of two months of computation. Overall, these results show that IBCD is competitive with the scalable baselines on real Perturb-seq data while additionally providing posterior edge uncertainty.
>
>
> We will also address the minor comments as follows. First, we will add the missing year to the Chickering citation. Second, we will clarify the notation in Section 3 and note that there was a typo ($x_i$ should be $y_i$) adding to the confusion. Let $\mathcal{I}$ denote the set of interventions, with $m∈\mathcal{I}$ indexing an intervention with possibly unknown targets. $p_m (y_i)$ denotes the marginal distribution of $y_i$ under intervention $m$, while $p_m( y_i | y_{pa_G(i)})$ denotes its conditional distribution given its parents in graph $G$ under intervention $m$. In our experiments, $X_i$ is the perturbation indicator for the intervention targeting $Y_i$. Third, $λ_{ij}$ is the local horseshoe scale parameter controlling edge-specific shrinkage (eq 20). Fourth, we will revise Section 4.3 so that the expected metric analysis and Table 7 are moved together into the main text. Fifth, BayesDAG and LLCB are the Bayesian baselines, whereas GES and GIES do not produce posterior samples. For those methods, expected metrics require bootstrap resampling rather than posterior samples as noted above.
>
> To clarify, IBCD does not assume that the exogenous noise terms are Gaussian. In Section 3.1, we assume independent exogenous variables with bounded variance, but do not otherwise specify their distributions. The Gaussian approximation enters at a different place where we use the asymptotic normality of the 2SLS total-effect estimator $\hat{R}$ to construct a tractable matrix-normal likelihood.
>
> To check this approximation in finite sample non-Gaussian settings, we ran a Poisson count-data simulation with 10,000 control samples and 100 intervention samples. Specifically, the exposure variable was generated as Y1 ~ Poisson(10) in control samples and Y1~ Poisson(3.6) in intervention samples, representing a perturbation-induced decrease in the exposure. The outcome variable was then generated from a Poisson model with conditional mean E[Y2 | Y1] = exp(log(10) + 0.3 log(Y1)). We then estimated the causal effect using 2SLS, repeated this procedure 1,000 times, and performed KS tests against normality. The tests consistently failed to reject the normality null; across repeated experiments, the p-value distribution was approximately flat, with median 0.573 and IQR 0.313–0.790. We will clarify this distinction and include this diagnostic with a Q-Q plot in the appendix.

---

### Review · Reviewer_DbGD · 2026-06-26

**Summary Of Contributions:**

The paper proposes an empirical Bayesian method for learning causal graphs from large-scale perturbation data. It first estimates total causal effects from interventions, and then recovers direct edges through the linear relation between the total-effect matrix and the adjacency matrix. I think the idea is useful, especially for Perturb-seq style data where known interventions are available. My main concern is about positioning. The paper mixes effect estimation, causal discovery, Bayesian inference, and single-cell perturbation analysis, but does not clearly explain the exact regime it is solving.

**Additional Comments:**

Overall, I like the direction. The paper would be stronger if it framed itself more cleanly as scalable Bayesian graph recovery from noisy total-effect estimates under a linear complete-intervention model. This is a useful contribution, but the current framing makes the scope look broader than it really is.

**Audience:**

Yes

**Audience Explanation:**

This should be interesting to people working on causal discovery, Bayesian structure learning, and single-cell perturbation data. The paper is in a useful regime: large known-target perturbation data, where purely observational causal discovery is not the right comparison and where direct Bayesian graph inference is usually too expensive. I think the paper would be more useful to the audience if it clearly explained how it differs from standard causal discovery, from effect estimation, and from perturbation prediction / representation learning.

**Broader Impact Concerns:**

No major concerns. The main issue is that inferred biological edges should be treated as hypotheses, not validated causal mechanisms.

**Claims And Evidence:**

Yes

**Claims Explanation:**

Mostly yes, but within a fairly strong regime. The synthetic experiments support the method under a linear SEM with known interventions and full perturbation coverage. I would not view this as a new identifiability result; the main contribution is more about scalable finite-sample inference from noisy total-effect estimates. For real data, the results show PIP stability/reproducibility, but not ground-truth correctness of direct biological causal edges. The wording should be a bit more careful here.

**Requested Changes:**

The main thing I would like to see is a clearer related work / positioning section. The method is really mixing effect estimation and causal discovery: first estimating total effects from complete known-target interventions, and then using these estimates to recover direct edges. I think this regime should be stated much more explicitly.

The related work also feels a bit thin for a causal discovery paper. The authors should at least briefly discuss classical causal discovery methods, e.g., PC/FCI and score-matching-based discovery methods. I do not think all of them need to be baselines, but the taxonomy should be clearer. The current baseline discussion also mixes observational and interventional methods. NOTEARS, GES, and LiNGAM are useful references, but they do not use the same information as IBCD. The main comparisons should really be with methods that use interventions or perturbation effects. Relatedly, there are interventional NOTEARS-style / differentiable causal discovery methods that seem relevant. If these methods are omitted because they assume hard interventions, do not scale, or do not fit this setting, this should be stated.

Since the paper is about single-cell perturbation data, I also think it should mention recent work on perturbation prediction / representation learning, e.g., Jiang et al., “What Makes a Good Representation for Single-Cell Perturbation Prediction?”, ICML 2026. This is not a direct baseline, but it is relevant context.

Method-wise, I have a few questions. First, the whole pipeline depends on the 2SLS/IV step giving meaningful total-effect estimates. In Perturb-seq data, the IV assumptions are not automatic: off-target effects, cell-state shifts, weak perturbations, or toxicity could all violate the exclusion restriction. Second, the matrix-normal likelihood is central to both scalability and PIP uncertainty, so I would like to see more justification or diagnostics for this approximation. Third, since the empirical Bayes prior is learned from summaries related to the same data used in the likelihood, the interpretation of PIPs should probably be more cautious.

Finally, I think the assumptions should be presented more clearly as part of the scope/limitations. The method assumes linearity, known targets, valid instruments, and essentially full perturbation coverage. These assumptions are fine for defining a specific regime, but the paper should be careful not to make the contribution sound broader than it is.

---

> ### Author Response · Authors · 2026-07-08
>
> Thank you for the thoughtful comments. We agree that the paper would be clearer if the exact assumptions made were stated earlier on, we will add these to the beginning of Section 3. We will also clarify where IBCD fits in the taxonomy of causal discovery methods including classical constraint/score based methods; differentiable methods; Bayesian methods; perturbation prediction work; and single cell representation learning (including the linked paper).
>
> We agree that the exact relationship between perturbation indicators $X$, variables $Y$, and IV assumptions should be made clearer in Section 3. Further, we will add this discussion of the IV assumptions in the appendix:
>
> In the simplest case, we assume $M=D$ instruments, with one instrument $X_i$​ for each variable $Y_i$​. That is, for each variable, $Y_i=X_i β_i​+∑​G_{ji}​Y_j​+ϵ_i$​. This encodes the three core IV assumptions; relevance, $X_i$ is associated with the exposure $Y_i$​; exchangeability, there is no unobserved common cause of $X_i$ and the outcome except through the modeled system; and exclusion restriction, $X_i$ affects $Y_j$ for $j≠i$  only through the effect of $Y_i$ on $Y_j$​. Under this model, the total causal effect $R_ij$ is estimated by 2SLS using $X_i$​ as the instrument, $Y_i$ as the treatment, and $Y_j$ as the outcome. Relevance is supported because the guide target is known and we filter genes whose guide does not have an effect on the target gene. Exchangeability is plausible because the confounders of concern, such as batch and cell cycle effects, are not causes of guide assignment and are regressed out in preprocessing. Exclusion restriction is supported because the guide library used has been explicitly designed to minimize off target activity (Replogle et al. 2022; Doench et al 2016), with recent work by Hartman et al 2026 identifying <1% of guides as showing substantial off target effects. Lastly, our simulations and real data analysis use the one targeted intervention variable per gene setting $M=D$, but it can allow $M>D$ when each variable has a nonempty set of valid instruments.
>
> We agree that the matrix normal likelihood should be presented as a structured approximation rather than an exact factorization. The paper currently motivates this by the asymptotic normality of the 2SLS total effect estimator and by the approximately separable row/column covariance structure, reducing the covariance burden from $D^4$ entries to $2D^2$. The nonseparable term is the residual covariance contribution Cov$(\hat{ϵ}\_{i​j}, \hat{ϵ}\_{kl}​ )$, where i,k index interventions and j,l index genes. Analytically, this decomposition is exact when $R_{ij}=R_{kl}=0$. To assess the approximation away from this case, we computed the residual covariance contribution for each intervention and compared it to the intervention averaged quantity used in the approximation. Using zero effect cases as a background baseline, we found that across 10 simulated D=50 datasets, the 2SD outlier rate among true effect cases was 16.87±1.70% for ER graphs and 42.12±4.15% for SF graphs; across all covariance matrix entries, the corresponding rates were 9.50±1.30% and 8.29±0.79%. We clarify that this is empirically reasonable for the studied regime.
>
> We agree that PIPs should be interpreted carefully and will revise the wording so that PIPs are described as posterior edge support scores under the empirical Bayes model. Importantly, the observational covariance summaries are used only to localize edge-specific prior weights, whereas the likelihood is based on the intervention derived total effect estimates $\hat{R}$. The paper evaluates whether PIPs are stable by using posterior calibration on D=500 simulations and by checking cross validation agreement and cross screen agreement on 521 gene Perturb-seq data.
>
> We tested how posterior uncertainty changes with intervention sample size to show the robustness of PIPs. Using the same D=50 intervention sample size experiment in Figure 3, we quantified posterior uncertainty using the mean over edges of min⁡(PIP,1−PIP), where smaller values indicate that posterior edge probabilities are more concentrated near 0 or 1 and are therefore less uncertain. Posterior uncertainty decreased as the number of intervention samples increased:
>
> | Graph | 5 | 15 | 25 | 50 | 75 | 100 |
> |---|---:|---:|---:|---:|---:|---:|
> | ER | 0.138±0.011 | 0.161±0.018 | 0.168±0.017 | 0.143±0.008 | 0.121±0.006 | 0.103±0.007 |
> | SF | 0.080±0.028 | 0.070±0.013 | 0.051±0.005 | 0.029±0.003 | 0.022±0.003 | 0.016±0.002 |
>
> Thus as intervention sample size increases, the posterior becomes more concentrated and less sensitive, consistent with the performance trends shown in Figure 3.
>
> Finally, we agree with the reviewer that real data biological edges should be treated as hypotheses, not as validated causal mechanisms. We will revise the real data section and discussion to say that IBCD prioritizes high confidence candidate direct effects for follow up experiments.

---

> > ### Author Response · Authors · 2026-07-08
> >
> > References:
> >
> > Replogle, J.M. et al. Mapping information-rich genotype-phenotype landscapes with genome-scale perturb-seq. Cell, 185(14):2559–2575, 2022.
> >
> > Doench, J. G. et al. Optimized sgRNA design to maximize activity and minimize off-target effects of CRISPR-Cas9. Nat. Biotechnol. 34, 184–191 (2016).
> >
> > Hartman A. et al. Systematic identification of seed-driven off-target effects in Perturb-seq experiments. bioRxiv 2026.03.27.714658

---

### Review · Reviewer_gDhF · 2026-06-27

**Summary Of Contributions:**

1. This paper proposes an empirical Bayesian method for causal discovery from intervention data.

2. This method assumes soft interventions, which is more realistic for CRISPR perturbations.

3. This method can scale up to solve large problem.

4. This method can give uncertainty for each edge.

**Audience:**

Yes

**Audience Explanation:**

This paper studies causal discovery, Bayesian inference, uncertainty quantification from intervention data. These topics are relevant to machine learning researchers, especially those working on causality, probabilistic modeling, and scientific applications such as genomics.

**Broader Impact Concerns:**

No concerns.

**Claims And Evidence:**

Yes

**Claims Explanation:**

Overall, this paper provides both theoretical and experimental results to support its claims. The evidence is convincing and clearly presented. However, some revisions are still needed before the paper can be accepted (see Requested Changes).

**Requested Changes:**

1. Is the model used in this paper acyclic or cyclic? The first sentence of the abstract mentions a “directed acyclic graph,” and the Introduction also states, as the first contribution, that the method aims at “inferring the DAG.” However, the proposed model appears to allow cycles. In particular, the paper does not impose an acyclicity constraint when defining the model, and Appendix G states that “IBCD does not require the inferred graph to be acyclic.” This inconsistency is important. For example, if the model is a DAG, one can write $R=(I-G)^{-1}=I+G+G^2+\cdots$ as in Equations (5) and (6). However, if the model is cyclic, additional conditions or assumptions are needed to ensure that this expression is well-defined. The authors should clarify this point and make the analysis more rigorous.

2. Typo in Equation (5): the authors seem to mean $R_{ij}=\sum_k R_{ik}G_{kj}$, but it should be $R_{ij}=\delta_{ij}+\sum_k R_{ik}G_{kj}$.

3. In the Introduction, the authors claim that IBCD is “the first causal discovery method that scales to large problems while quantifying uncertainty.” However, methods such as BayesDAG and BCD Nets also appear to scale to relatively large problems while providing uncertainty quantification.

---

> ### Author Response · Authors · 2026-07-08
>
> We thank the reviewer for the thoughtful comments.
>
> Regarding the first question, we agree that the current wording should more clearly distinguish the DAG assumption from the inference procedure. IBCD assumes a DAG data generating model, and the synthetic ground truth graphs are DAGs. The total effect follows the linear DAG SEM where $R = (I - G)^{-1}$, equivalently $G = I- R^{-1}$. The point of Appendix G is not that the model targets cyclic graphs, but that posterior sampling does not impose a hard acyclicity constraint on every sampled estimate of G. We will make sure to update this point in the text.
>
> We thank the reviewer for pointing out the typo in eq (5). Indeed, as written $R_{ii}$ is undefined as $G_{ii} := 0$, when instead $R_{ii}$ should be 1, as indicated by $\delta_{ij}$. We will update the equation accordingly.
>
> Lastly, we agree that the original wording was too broad. Our intended claim is not that no prior Bayesian causal discovery method provides uncertainty, but that IBCD scales to the hundreds of variables while producing edge level posterior inclusion probabilities. In our experiments, Bayesian baselines, BayesDAG and LLCB, exceeded practical runtime or memory limits beyond D=50 (Table 1), and our initial attempts to run BCD Nets were not successful in this setting. By contrast, IBCD ran up to D=500 in simulation and was applied to 521 genes in Perturb-seq data. We will revise the wording to make this empirical scope clear and acknowledge related Bayesian approaches more carefully.